# Sumoylation promotes optimal APC/C activation and timely anaphase

**Christine C Lee[1], Bing Li[2], Hongtao Yu[2], Michael J Matunis[1]\***

[1]Department of Biochemistry and Molecular Biology, Johns Hopkins University, Baltimore, United States; [2]Department of Pharmacology, Howard Hughes Medical Institute, University of Texas Southwestern Medical Center, Dallas, United States

**Abstract** The Anaphase Promoting Complex/Cyclosome (APC/C) is a ubiquitin E3 ligase that functions as the gatekeeper to mitotic exit. APC/C activity is controlled by an interplay of multiple pathways during mitosis, including the spindle assembly checkpoint (SAC), that are not yet fully understood. Here, we show that sumoylation of the APC4 subunit of the APC/C peaks during mitosis and is critical for timely APC/C activation and anaphase onset. We have also identified a functionally important SUMO interacting motif in the cullin-homology domain of APC2 located near the APC4 sumoylation sites and APC/C catalytic core. Our findings provide evidence of an important regulatory role for SUMO modification and binding in affecting APC/C activation and mitotic exit.

DOI: https://doi.org/10.7554/eLife.29539.001

## Introduction

Aberrations during cell division can result in abnormal chromosome number or aneuploidy, a key hallmark of human cancers (*Kops, 2014*). Cellular mechanisms therefore exist to safeguard against chromosome missegregation, including the spindle assembly checkpoint (SAC) that functions during the mitotic stage of the cell cycle. At the onset of mitosis, mitotic kinases and microtubule motor proteins coordinate attachment of spindle microtubules to kinetochores to align sister chromatids at the metaphase plate (*Foley and Kapoor, 2013*). The SAC monitors spindle attachments and produces an inhibitory signal known as the Mitotic Checkpoint Complex (MCC) (*Lara-Gonzalez et al., 2012*; *Musacchio and Salmon, 2007*). The MCC is composed of BubR1, Bub3, Cdc20, and Mad2 and functions to inhibit the Anaphase Promoting Complex or Cyclosome (APC/C), a ubiquitin RING E3 ligase (*Musacchio, 2015*). The SAC is silenced upon proper chromosome alignment, leading to APC/C activation, processive ubiquitylation and degradation of Cyclin B1 and securin, and subsequent anaphase onset (*Sivakumar and Gorbsky, 2015*). The dynamics between SAC signaling and APC/C activation produce a fine-tuned 'rheostat' responsive to multiple signals affecting mitotic exit (*Collin et al., 2013*).

The multi-subunit APC/C is a member in a large family of cullin-RING E3 ligases and functions most prominently as the gatekeeper to the metaphase-anaphase transition and mitotic exit. APC/C activity must be exquisitely regulated to target specific substrates for ubiquitin-mediated proteolysis during different points in the cell cycle (*Alfieri et al., 2017*; *Pines, 2011*). In mitosis, APC/C activity is modulated through multiple signals that act to regulate MCC, co-activator, E2 enzyme and substrate binding (*Eytan et al., 2014*; *Mansfeld et al., 2011*; *Uzunova et al., 2012*; *Matyskiela and Morgan, 2009*), as well as sub-cellular localization (*Acquaviva et al., 2004*; *Jörgensen et al., 1998*; *Topper et al., 2002*). Studies of APC/C structure (*Alfieri et al., 2016*; *Brown et al., 2014*; *Chang et al., 2014*; *Herzog et al., 2009*; *Yamaguchi et al., 2016*) and interactions with E2 enzymes (*Brown et al., 2016*; *Kelly et al., 2014*) and substrates (*Davey and Morgan, 2016*) are beginning to provide detailed molecular insights into its function and regulation. These findings reveal that the

**\*For correspondence:**
mmatuni1@jhu.edu

**Competing interests:** The authors declare that no competing interests exist.

APC/C contains a flexible and dynamic catalytic core composed of APC2, the cullin subunit, and APC11 that contains the RING domain (*Alfieri et al., 2016*; *Brown et al., 2016*). Notably, the ability of the catalytic core to bind co-activators, substrates and E2 conjugating enzymes is directly impacted by MCC interactions. Although phosphorylation has been identified as an important regulator of co-activator binding (*Alfieri et al., 2016*; *Fujimitsu et al., 2016*; *Kraft et al., 2003*; *Yamaguchi et al., 2016*), mechanisms modulating MCC binding and dissociation from the APC/C remain incompletely understood.

Sumoylation regulates a variety of essential cellular processes, including DNA replication and repair, chromatin remodeling, and mitosis (*Flotho and Melchior, 2013*). Analogous to the ubiquitin pathway, SUMOs are added to proteins by the sequential cascade of E1, E2, and E3 enzymes. However, the catalog of SUMO pathway enzymes in vertebrates is more limited in comparison to the ubiquitin system, with a single activating E1 (expressed as the heterodimer Aos1/Uba2), a single E2 conjugating enzyme (Ubc9) and approximately a dozen E3 ligases (*Gareau and Lima, 2010*). The SUMO pathway also includes a family of cysteine isopeptidases, known as Sentrin isopeptidases (SENPs), that remove SUMOs from substrates (*Mukhopadhyay and Dasso, 2007*). Regulation at the level of deconjugation is an important axis controlling SUMO function, as demonstrated by the importance of SENP1 and SENP2 in regulating mitotic progression (*Cubeñas-Potts et al., 2013*). Although dynamic cycling between conjugation and de-conjugation can result in a relatively low steady-state level of sumoylation for most substrates, sumoylation nonetheless produces essential regulatory effects on substrate localization and function in a variety of cellular pathways.

One paradigm for SUMO function is its role as 'molecular Velcro' and in the generation of non-covalent interactions between proteins to facilitate protein complex assembly (*Jentsch and Psakhye, 2013*; *Matunis et al., 2006*). Novel protein-protein interactions are generated between sumoylated proteins and proteins containing SUMO-interacting motifs (SIMs). SUMO-SIM interactions are required for the assembly of cellular structures, including PML nuclear bodies (*Shen et al., 2006*), ribosomes (*Finkbeiner et al., 2011*) and kinetochores (*Mukhopadhyay et al., 2010*; *Shen et al., 2006*; *Zhang et al., 2008*). Early genetic studies in *S. cerevisiae* have also identified SUMO pathway components as essential for progression through mitosis (*Biggins et al., 2001*; *Meluh and Koshland, 1995*; *Seufert et al., 1995*). Yeast sumoylation mutants result in a large budded phenotype and fail to properly degrade APC/C substrates Pds1 and Clb3, indicating an essential role for sumoylation during the metaphase-anaphase transition (*Dieckhoff et al., 2004*). Precisely how sumoylation regulates the metaphase-anaphase transition, however, has not been defined. Additional investigations in human cell lines have underscored the importance of sumoylation in early mitotic processes, including kinetochore-microtubule interactions (*Li et al., 2016*; *Zhang et al., 2008*), sister chromatid cohesion (*Azuma et al., 2003*; dharan et al., 2015), and checkpoint signaling (*Ban et al., 2011*; *Fernández-Miranda et al., 2010*; *Yang et al., 2012*). Recent proteomic studies have also identified sumoylation sites on subunits of the APC/C, suggesting possible effects of SUMO on APC/C function (*Cubeñas-Potts et al., 2015*; *Matic et al., 2010*; *Schimmel et al., 2014*; *Schou et al., 2014*).

Here, we demonstrate that the APC/C subunit APC4 is sumoylated at two C-terminal residues. We show that APC4 sumoylation peaks during mitosis and is required for timely anaphase onset that is SAC-dependent. In addition, our findings demonstrate that the APC2 subunit contains a functional SIM near the C-terminal winged-helix B (WHB) domain and in close spatial proximity to APC4 sumoylation sites. Analyses of APC4 and APC2 mutants in cultured mammalian cells are consistent with SUMO-SIM interactions between these subunits contributing to timely APC/C activation and mitotic exit.

## Results

### APC4 is sumoylated in a cell-cycle-dependent manner at lysines 772 and 798

APC4 is a protein subunit at the base of the APC/C that is sumoylated in mitosis (*Cubeñas-Potts et al., 2015*). To more precisely characterize the temporal regulation of APC4 sumoylation during the cell cycle, we synchronized HeLa cells using a double-thymidine block. After release from thymidine for varying lengths of time, cell lysates were analyzed by immunoblotting for APC4 and

the APC/C substrates Cyclin B1 and Cdc20 (*Figure 1A*). Consistent with a possible role in regulating APC/C activity in mitosis, APC4 sumoylation levels (evidenced by the detection of a prominent high-molecular-mass protein band migrating at 120 kDa) increased with entry into mitosis and peaked at a time correlating with Cyclin B1 degradation. APC4 sumoylation was further investigated using a

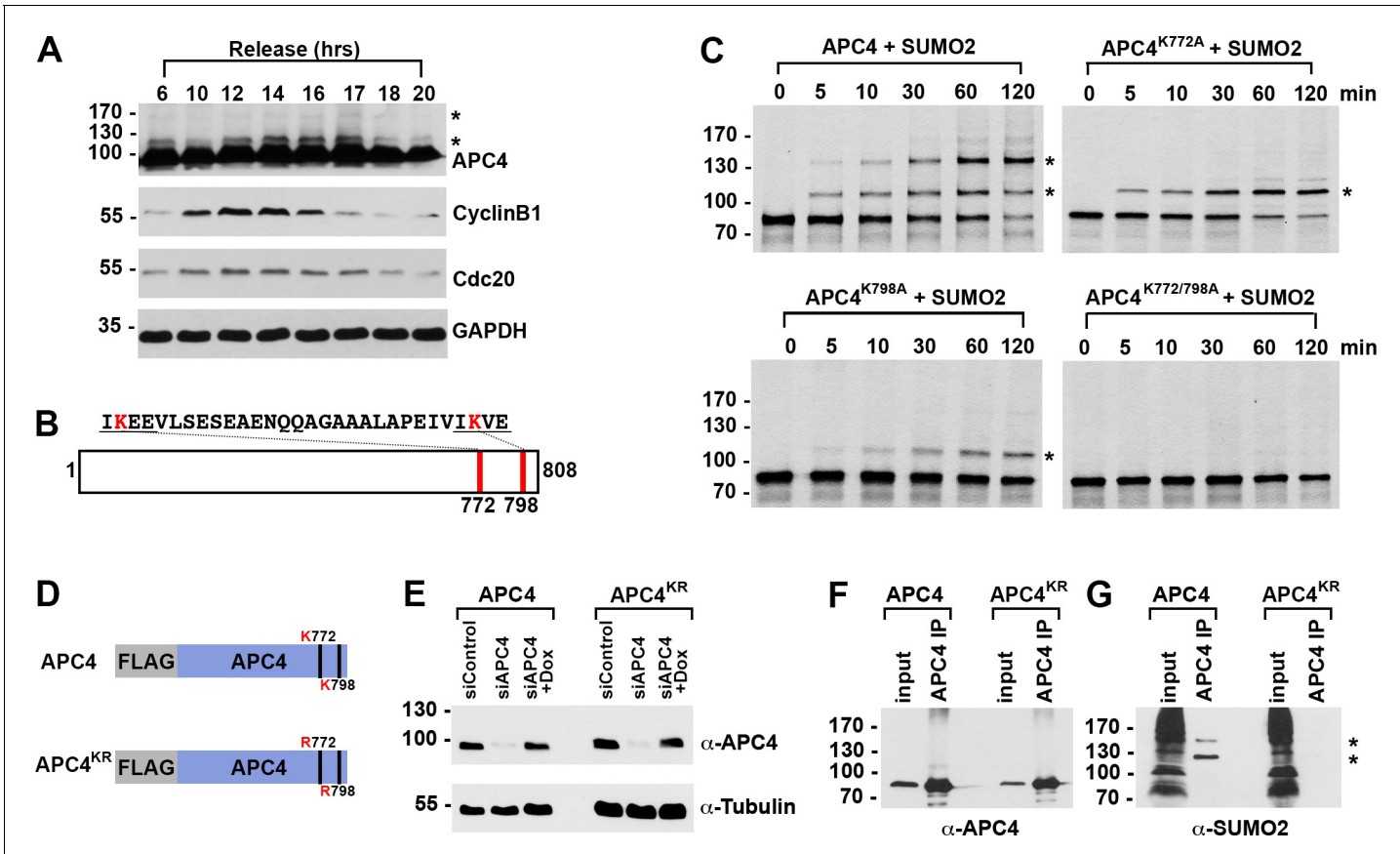

**Figure 1.** APC4 is sumoylated in a cell-cycle-dependent manner at two C-terminal lysines. (**A**) HeLa cells were synchronized in S-phase using a double-thymidine arrest and released for varying time points. Whole cell lysates were analyzed by immunoblotting for APC4, Cyclin B1, Cdc20, and glyceraldehyde 3-phosphate dehydrogenase (GAPDH) as a loading control. Asterisks indicate sumoylated forms of APC4. (**B**) APC4 contains two C-terminal SUMO consensus site lysines at 772 and 798. (**C**) Full-length wild-type APC4 or the indicated lysine to alanine substitution mutants were expressed in rabbit reticulocyte lysate in the presence of [$^{35}$S]-methionine and incubated for the indicated times in modification reactions containing SUMO E1 and E2 enzymes and SUMO2. Proteins were detected by SDS-PAGE and autoradiography. Asterisks indicate sumoylated forms of APC4. (**D**) Constructs coding for FLAG-tagged versions of wild type APC4 or a sumoylation-deficient mutant containing arginine substitutions at lysines 772 and 798 (APC4$^{KR}$) were used to generate stable inducible cell lines in YFP-H2B HeLa cells. (**E**) Endogenous APC4 was depleted by siRNA, and FLAG-APC4 or FLAG-APC4$^{KR}$ stable cell lines were induced by doxycycline for 48 hr. Immunoblot analysis using APC4 and tubulin-specific antibodies reveals that FLAG-APC4 and FLAG-APC4$^{KR}$ are expressed at near endogenous levels. (**F–G**) Co-immunoprecipitations were performed with an antibody against APC4, followed by immunoblotting for APC4 or SUMO2. FLAG-APC4 is sumoylated in vivo while FLAG-APC4$^{KR}$ is not. Asterisks indicate sumoylated APC4.

DOI: https://doi.org/10.7554/eLife.29539.002

The following video and figure supplement are available for figure 1:

**Figure supplement 1.** APC4 is sumoylated at two C-terminal lysine residues in mitosis; APC4 sumoylation is regulated by SENP1.

DOI: https://doi.org/10.7554/eLife.29539.003

**Figure 1—video 1.** Movie of YFP-H2B (green) HeLa cells treated for 48 hr with a control siRNA.

DOI: https://doi.org/10.7554/eLife.29539.004

**Figure 1—video 2.** Movie of YFP-H2B (green) HeLa cells treated for 48 hr with siAPC4.

DOI: https://doi.org/10.7554/eLife.29539.005

**Figure 1—video 3.** Movie of FLAG-APC4 WT cells treated for 48 hr with siAPC4 and APC4 WT induction with doxycycline.

DOI: https://doi.org/10.7554/eLife.29539.006

U2OS cell line stably expressing 6xHis-SUMO2 and the parent U2OS cell line as control. Cells were synchronized in S-phase using a double-thymidine block or in mitosis using nocodazole treatment and release. Proteins were captured from cell lysates using nickel-NTA agarose and analyzed by immunoblotting with anti-APC4 and SUMO2/3 antibodies (*Figure 1—figure supplement 1A*). Although unmodified APC4 (~97 kDa) was non-specifically purified from control and 6xHis-SUMO-2 expressing cell lysates, SUMO2-modified APC4 (~120 kDa) was uniquely identified in 6xHis-SUMO-2 expressing cells. Further supporting a function in mitosis, maximal levels of sumoylated APC4 were detected in nocodazole-arrested cells and levels decreased following exit from mitosis (*Figure 1— figure supplement 1A*).

Human APC4 contains two C-terminal lysine residues located within consensus sumoylation sites, at positions 772 and 798, that have previously been reported to be sumoylated in high throughput mass spectrometry studies (*Cubeñas-Potts et al., 2015*; *Matic et al., 2010*; *Schimmel et al., 2014*; *Schou et al., 2014*) (*Figure 1B*). These APC4 consensus sites are conserved across mammals (*Figure 2—figure supplement 1A*). To characterize APC4 sumoylation and modification at K772 and K798, we expressed wild type and mutant variants of APC4 in rabbit reticulocyte lysate in the presence of [$^{35}$S]-methionine. Translation products were incubated in reactions containing recombinant SUMO E1 activating and E2 conjugating enzymes and SUMO2 for varying lengths of time and analyzed by SDS-PAGE and autoradiography (*Figure 1C*). In reactions containing wild-type APC4, we observed two prominent high-molecular-mass bands of 120 and 135 kDa, consistent with sumoylation at two sites. In reactions containing APC4 with single amino acid substitutions at either K772A or K798A, a single high-molecular-mass band of 120 kDa was observed, whereas sumoylation was abolished in reactions containing the K772A/798A double mutant. Of note, lysine 798 was more efficiently modified relative to lysine 772. Reactions were also performed in the presence of recombinant SUMO1 with comparable results (*Figure 1—figure supplement 1B*). Thus, our findings are consistent with previous mass spectrometry studies indicating that APC4 is sumoylated at two C-terminal lysine residues at positions 772 and 798.

To characterize sumoylation of APC4 in vivo, we generated stable HeLa cell lines allowing for inducible expression of FLAG-tagged, wild-type APC4 or a K772/798R mutant, hereafter referred to as APC4 and APC4$^{KR}$ (*Figure 1D*). These cells also constitutively expressed yellow fluorescent protein (YFP)-tagged histone H2B to facilitate live cell imaging. To investigate APC4 sumoylation using these cell lines, we depleted endogenous APC4 using siRNAs directed against the 3' UTR and induced expression of the FLAG-tagged transgenes to near endogenous levels (*Figure 1E*). APC4 was immunopurified from cell lysates using FLAG-specific antibodies and immunoblot analysis was performed using APC4- or SUMO2/3-specific antibodies. Two SUMO2/3-modified protein bands were detected at 120 and 135 kDa in immunopurifications from wild type cells (*Figure 1F*), but not in immunopurifications from cells expressing APC4$^{KR}$ (*Figure 1G*). We also evaluated sumoylation by SUMO1 with comparable results (*Figure 1—figure supplement 1C*). These findings confirm that K772 and K798 are the major APC4 sumoylation sites in vivo.

We also explored the regulation of APC4 sumoylation by SUMO isopeptidases using siRNA-mediated depletion of SENP1 or SENP2, two isopeptidases critical for normal mitotic progression (*Cubeñas-Potts et al., 2013*). SENP1 or SENP2 were depleted using siRNAs as previously described (*Cubeñas-Potts et al., 2013*), and cells were synchronized at metaphase using a nocodazole block followed by a 2 hr release into medium containing the proteasome inhibitor MG132. Immunoblot analysis of cell lysates revealed that APC4 sumoylation is specifically enhanced in SENP1-depleted cells compared to control or SENP2-depleted cells (*Figure 1—figure supplement 1D*), implicating SENP1 as a regulator of APC4 sumoylation in mitosis.

## Sumoylation of APC4 is required for timely metaphase-anaphase transition

As a subunit of the APC/C, APC4 is predicted to be essential for normal cell cycle progression. To test this prediction, we compared cells transfected with control and APC4 specific siRNAs and analyzed cells after 48 hr using timelapse microscopy (*Figure 1—videos 1* and *2*). This analysis revealed that ~ 88% of cells exhibit severe defects in mitosis (*Figure 2A*). Specifically, 20% of cells undergo a prolonged metaphase (as defined as >60 min) and 69% undergo an abnormal mitosis that includes defects in chromosome congression, an inability to maintain a metaphase plate and cell death, as

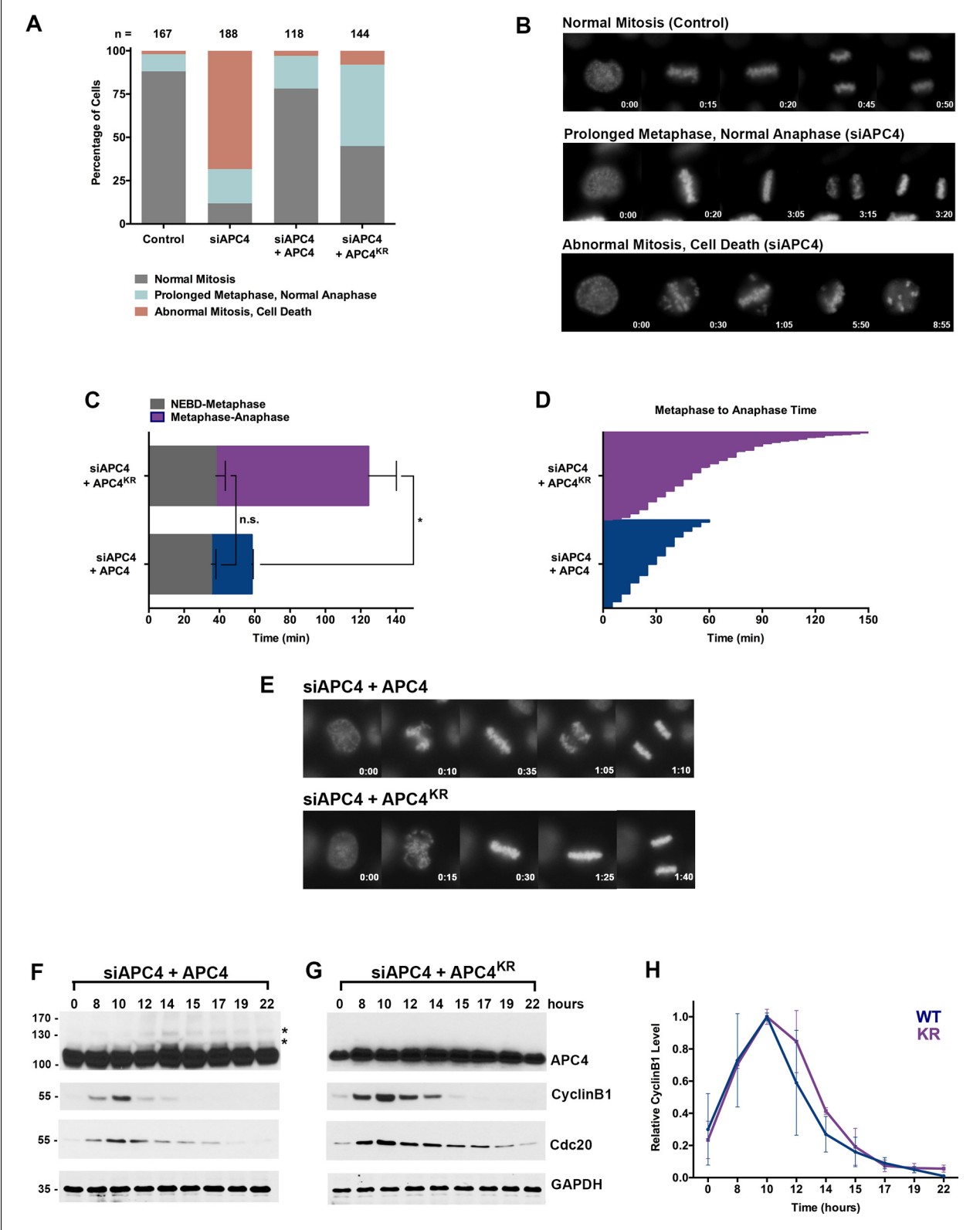

**Figure 2.** APC4 sumoylation is required for timely metaphase-anaphase transition. (**A**) Cells were transfected with control or APC4-specific siRNAs for 48 hr followed by 16 hr of timelapse live cell acquisition. siAPC4 transfected cells were also induced to express FLAG-APC4 or FLAG-APC4[KR]. Analysis represents mitotic progression time beginning with nuclear envelope breakdown (NEBD) to anaphase onset. Quantification of mitotic phenotypes is shown. Prolonged metaphase is defined by >60 min in metaphase plate alignment before anaphase onset. Abnormal metaphase is defined by inability

*Figure 2 continued on next page*

*Figure 2 continued*

to generate a metaphase plate and defects in chromosomal cohesion. n > 100 for each cell line. (B) Cells representative of each mitotic phenotype categorized in (A) are featured with timestamps in minutes. (C) Mitotic progression beginning with NEBD to metaphase plate alignment and from metaphase plate alignment to anaphase onset was quantified in FLAG-APC4 and FLAG-APC4$^{KR}$ expressing cells. Experiments were performed in triplicate; means are displayed and error bars represent standard deviations. n = 50 for each cell line. Two-tailed t-tests were used to calculate significance: p=0.38 for differences in FLAG-APC4 and FLAG-APC4$^{KR}$ timing from NEBD-metaphase, p<0.001 for differences in FLAG-APC4 and FLAG-APC4$^{KR}$ timing from metaphase plate alignment to anaphase. (D) Individual timing of metaphase-anaphase progression is displayed for FLAG-APC4 and FLAG-APC4$^{KR}$ expressing cells. n = 215 for each cell line. (E) Representative cells from timelapse acquisition beginning with NEBD to anaphase onset in FLAG-APC4 and FLAG-APC4$^{KR}$ expressing cells with timestamps indicated in minutes. (F) FLAG-APC4 and (G) FLAG-APC4$^{KR}$ expressing cells were synchronized in S-phase using a double-thymidine block and released for various time points. Whole cell lysates were analyzed by immunoblotting for APC4, Cyclin B1, Cdc20 and GAPDH as a loading control. Asterisks (*) indicate sumoylated forms of APC4. (H) Relative protein levels of Cyclin B1 in FLAG-APC4 and FLAG-APC4$^{KR}$ cells were quantitated using ImageJ. Normalized mean values are graphed with standard deviations from three separate experiments.

DOI: https://doi.org/10.7554/eLife.29539.007

The following video and figure supplement are available for figure 2:

**Figure supplement 1.** Analysis of single SUMO consensus site APC4 mutants.

DOI: https://doi.org/10.7554/eLife.29539.008

**Figure 2—video 1.** Movie of FLAG-APC4$^{KR}$ cells treated for 48 hr with siAPC4 and APC4 KR induction with doxycycline.

DOI: https://doi.org/10.7554/eLife.29539.009

**Figure 2—video 2.** Movie of FLAG-APC4$^{K772R}$ cells treated for 48 hr with siAPC4 and APC4$^{K772R}$ induction with doxycycline.

DOI: https://doi.org/10.7554/eLife.29539.010

**Figure 2—video 3.** Movie of FLAG-APC4$^{K798R}$ cells treated for 48 hr with siAPC4 and APC4$^{K798R}$ induction with doxycycline.

DOI: https://doi.org/10.7554/eLife.29539.011

depicted in single frame images (*Figure 2B*). These observations are consistent with defects observed in cells depleted of other essential APC/C subunits (*de Lange et al., 2015*).

Having established that APC4 is an essential APC/C subunit required for mitotic progression, we next sought to investigate the function of APC4 sumoylation with knockdown and rescue experiments using the inducible cell lines described above. Cells were depleted of endogenous APC4 by siRNA-mediated knockdown and APC4 or APC4$^{KR}$ transgenes were concomitantly induced for 48 hr. Progression through mitosis was analyzed after 16 hr of timelapse acquisition beginning with nuclear envelope breakdown (NEBD) and ending with anaphase onset (*Figure 1—video 3* and *Figure 2—video 1*). Notably, expression of the APC4$^{KR}$ mutant was not fully effective in rescuing the APC4 knockdown defects, in comparison to full rescue by APC4 (*Figure 2A and C–E*). In particular, analysis revealed that cells expressing APC4 or APC4$^{KR}$ did not have statistically significant differences (p=0.38) in the time from NEBD to metaphase, but exhibited significant differences (p<0.0001) in the time between metaphase and anaphase onset (*Figure 2C–D*). On average, APC4-expressing cells spent ~23 min from metaphase to anaphase and APC4$^{KR}$ cells spent ~86 min. Despite the observed delay, APC4$^{KR}$ expressing cells ultimately progressed to anaphase and exited mitosis.

We also characterized the effects of individual APC4 consensus sumoylation site mutations on mitotic progression by analysis of inducible cell lines expressing K772R or K798R APC4 mutants (*Figure 2—figure supplement 1 B and C*). Cells were depleted of endogenous APC4 using siRNA and FLAG-APC4$^{772R}$ (APC4$^{772R}$) or FLAG-APC4$^{798R}$ (APC4$^{798R}$) expression was induced using doxycycline for 48 hr prior to timelapse acquisition (*Figure 2—videos 2 and 3*). Most notably, APC4$^{K772R}$ and APC4$^{K798R}$ expressing cells exhibited metaphase-anaphase delays that were intermediate to APC4$^{KR}$ double-mutant expressing cells (*Figure 2—figure supplement 1D–F*). On average, APC4$^{772R}$ cells took ~40 min to transition from metaphase to anaphase onset and APC4$^{798R}$ took ~65 min, compared to ~86 min in APC4$^{KR}$ double-mutant expressing cells and ~26 min in control cells (*Figure 2—figure supplement 1D–F*, *Figure 2C–E*). Thus, sumoylation at a single site is sufficient (798 > 772) to partially restore APC4 function.

To explore the molecular basis of the metaphase-anaphase delay observed in APC4$^{KR}$ expressing cells, we monitored the degradation of the APC/C substrates, Cyclin B1 and Cdc20. Following depletion of endogenous APC4 and induction of APC4 or APC4$^{KR}$, cells were synchronized using a double-thymidine block and released into drug-free medium. In APC4-expressing cells, APC4 sumoylation peaked concomitantly with declines in Cyclin B1 and Cdc20 protein levels, beginning at ~12 hr following thymidine release (*Figure 2F*). As expected, APC4 sumoylation was not observed

in APC4$^{KR}$-expressing cells (*Figure 2G*). In addition, the turnover of Cyclin B1 and Cdc20 protein levels was delayed compared to APC4-expressing cells. Specifically, in APC4-expressing cells, Cyclin B1 levels were reduced to 50% of peak levels approximately 12.5 hr following thymidine release, while in APC4$^{KR}$ expressing cells levels were reduced to 50% after 13.5 hr (*Figure 2F–H*). Thus, the delay in anaphase onset detected in APC4$^{KR}$-expressing cells correlates with a delay in the turnover of APC/C-dependent target proteins. Taken together, our findings suggest that APC4 sumoylation is required for timely APC/C activation or optimal function.

## Fusion of SUMO to APC4$^{KR}$ rescues the metaphase-anaphase delay

Having established that a defect in APC4 sumoylation affects timely anaphase onset, we next sought to investigate the consequences of constitutive APC4 sumoylation. Toward this end, we generated a stable cell line allowing for inducible expression of a FLAG-tagged linear APC4$^{KR}$-SUMO2 fusion protein (*Figure 3A*). Linear SUMO fusions have successfully mimicked conjugation at internal lysines in multiple other proteins (*Holmstrom et al., 2003*; *Yurchenko et al., 2006*). Our rationale for utilizing SUMO2 was based on in vitro analysis demonstrating that SUMO2 is conjugated to APC4 more efficiently (*Figure 1C*, *Figure 1—figure supplement 1B*) in addition to the finding that SUMO2/3 plays a dominant role during mitosis compared to SUMO1 (*Zhang et al., 2008*). A single SUMO2 (lacking its C-terminal di-glycine motif to prevent conjugation) was added to the C-terminus of APC4$^{KR}$ based on the observation that modification at a single lysine partially rescues function (*Figure 2—figure supplement 1E,F*).

To investigate the consequences of constitutive APC4 sumoylation, endogenous APC4 was depleted using siRNA knockdown and the APC4$^{KR}$-SUMO2 fusion protein (APC4$^{KR-S2}$ or KR-S2) was induced to near endogenous expression levels (*Figure 3B*). Following 48 hr of siRNA depletion and transgene expression, cell cycle progression was analyzed by timelapse microscopy (*Figure 3C*, *Figure 3—video 1*). Time course analysis revealed no measurable differences between control and APC4$^{KR-S2}$-expressing cells. No defects in the timing from NEBD to chromosome alignment at the metaphase plate were observed, and importantly, the timing from metaphase alignment to anaphase onset was normal (*Figure 3D–E*). On average, control cells took 23 min to progress from NEBD to metaphase and 27 min from metaphase to anaphase, while APC4$^{KR-S2}$ cells took 25 min to progress from NEBD to metaphase and 25 min from metaphase to anaphase. Thus, fusing SUMO2 to the C-terminus of APC4$^{KR}$ rescued the requirement for sumoylation at K772 and K798, a demonstration of the functionality of the linear fusion protein. The absence of any overt consequences on cell cycle progression reveals that constitutive sumoylation of APC4 is insufficient to induce untimely APC/C activation.

## APC4 sumoylation functions through the SAC but does not directly affect APC/C activity

The delayed turnover of both Cyclin B1 and Cdc20 in APC4$^{KR}$-expressing cells suggests that effects of sumoylation on APC/C may be linked to SAC silencing. To explore this further, we monitored the effect of either weakening or abrogating SAC signaling by treating cells with low and high doses of reversine, an inhibitor of the Mps1 kinase (*Santaguida et al., 2010*). Cells were depleted of endogenous APC4, and APC4 and APC4$^{KR}$ transgenes were induced for 48 hr followed by treatment with either 1 µM or 50 nM reversine (*Figure 4—videos 1–4*). Progression through mitosis, beginning with NEBD to anaphase onset, was analyzed after 4 hr of timelapse acquisition. When the SAC was fully inhibited with 1 µM reversine, APC4 and APC4$^{KR}$-expressing cells entered mitosis and rapidly exited with similar kinetics and chromosome segregation defects (~29 and 24 min, respectively) (*Figure 4A–H*). Thus, SAC abrogation eliminated the metaphase delay observed in APC4$^{KR}$-expressing cells and APC/C activity appeared to be similar in both APC4 and APC4$^{KR}$-expressing cells in the absence of the SAC. In contrast, weakening the SAC by treatment with a low dose of reversine elicited unique effects on APC4 and APC4$^{KR}$-expressing cells. Whereas the time in mitosis was significantly shortened in APC4-expressing cells treated with 50 nM reversine (from ~49 min in control to 35 min in treated), no significant effect was observed on mitotic progression in APC4$^{KR}$-expressing cells, which continued to exhibit a marked delay in the metaphase-anaphase transition (*Figure 4A–H*). Collectively, these findings are consistent with APC4 sumoylation affecting APC/C regulation by the SAC and activation downstream of SAC silencing.

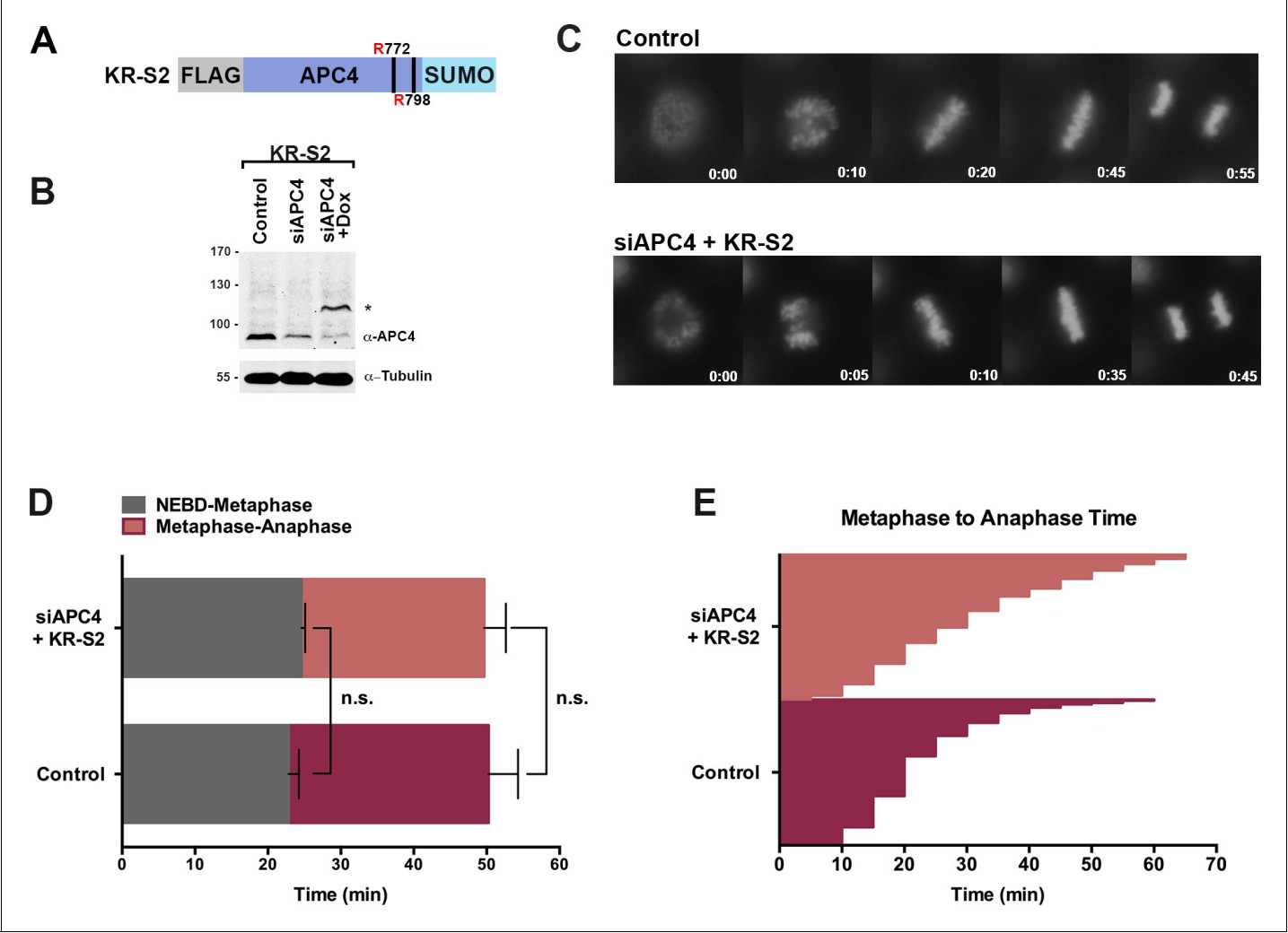

**Figure 3.** APC4^KR-SUMO rescues APC4^KR metaphase-anaphase delay. (**A**) A construct coding for a FLAG-tagged APC4^KR-SUMO2 fusion protein (APC4^KR-S2) was used to generate stable inducible lines in YFP-H2B HeLa cells. (**B**) Cells were transfected with control or APC4-targeting siRNAs and cultured in the presence (+Dox) or absence of doxycycline. Endogenous APC4 and FLAG-APC4^KR-S2 expression levels were analyzed by immunoblot analysis with APC4 and tubulin specific antibodies. (**C**) Representative cells from timelapse acquisition beginning with NEBD to anaphase onset in control and FLAG-APC4^KR-S2 expressing cells with timestamps indicated in minutes. (**D**) Mitotic timing beginning with NEBD to metaphase plate alignment and from metaphase plate alignment to anaphase onset was quantified in control and FLAG-APC4^KR-S2-expressing cells. Experiments were performed in triplicate; means are displayed and error bars represent standard deviations. n = 60 for each cell line. Two-tailed t-tests were used to calculate significance: p=0.16 between control and FLAG-APC4^KR-S2 timing from NEBD-metaphase, p=0.35 for control and FLAG-APC4^KR-S2 timing from metaphase plate alignment to anaphase. (**E**) Individual metaphase to anaphase transition times are displayed for control and FLAG-APC4^KR-S2-expressing cells. n = 314 for each cell line.

DOI: https://doi.org/10.7554/eLife.29539.012
The following video is available for figure 3:
**Figure 3—video 1.** Movie of FLAG-APC4^KR-S2 cells treated for 48 hr with siAPC4 and APC4^KR-S2 induction with doxycycline.
DOI: https://doi.org/10.7554/eLife.29539.013

To investigate whether APC4 sumoylation has a direct effect on APC/C activity, we performed in vitro ubiquitylation assays. APC/C was immunopurified from mitotic cell lysates and APC4 was sumoylated by incubating purified complexes with recombinant SUMO, E1 and E2 enzymes. Sumoylation of APC4 was efficient and specific, as revealed by immunoblot analysis with APC4 and SUMO-specific antibodies (*Figure 4J–K*). Ubiquitylation assays using SUMO-modified APC/C and unmodified control complexes, and the securin substrate, revealed indistinguishable activities (*Figure 4I*). Assays were also performed in the presence of inhibitory MCC complexes to investigate if

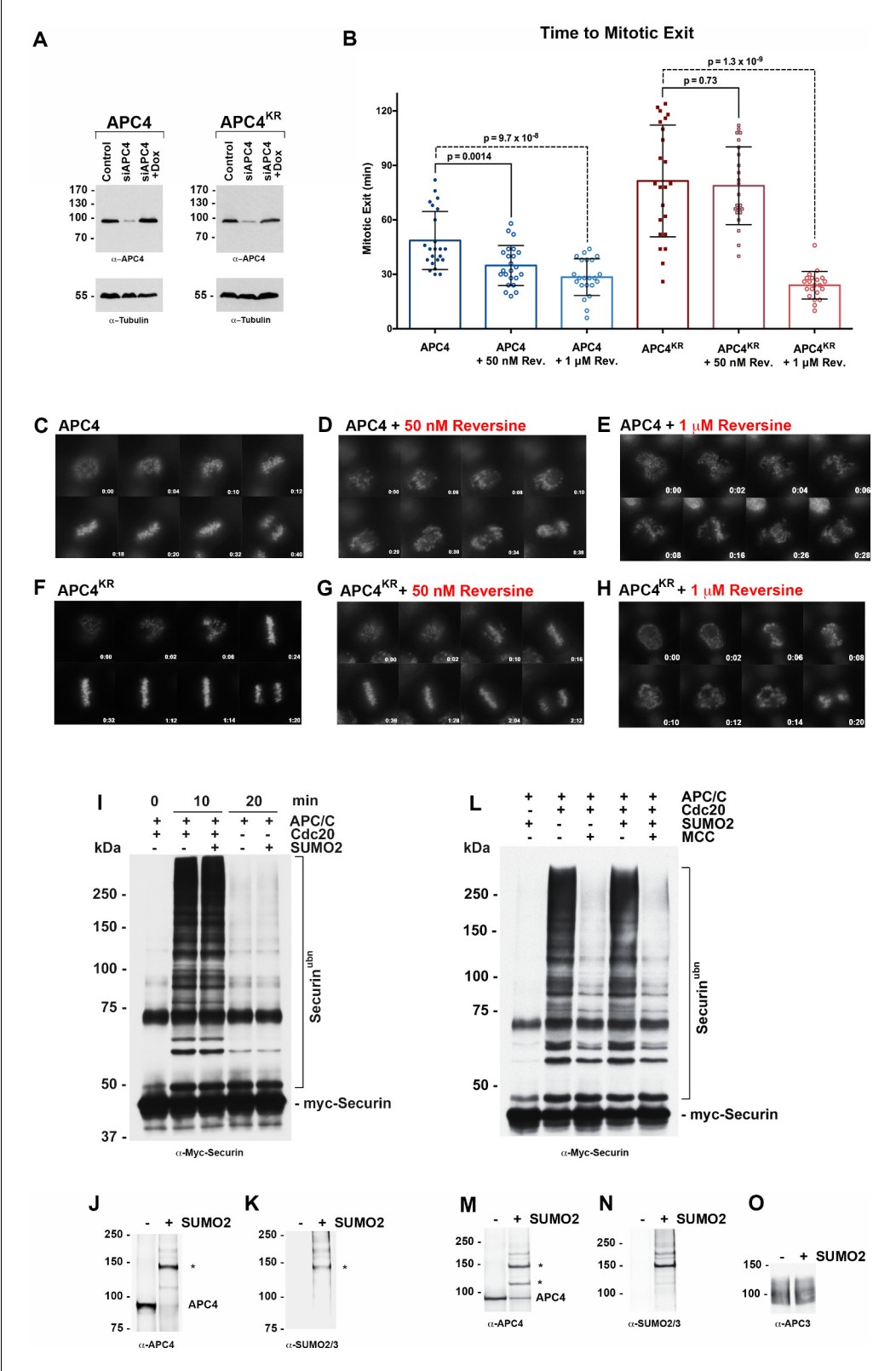

**Figure 4.** APC4 sumoylation indirectly affects APC/C activity and functions through the SAC. (**A**) Endogenous APC4 was depleted by siRNA, and FLAG-APC4 or FLAG-APC4^KR stable cell lines were induced by doxycycline for 48 hr. Immunoblot analysis using APC4 and tubulin-specific antibodies reveals that FLAG-APC4 and FLAG-APC4^KR are expressed at near endogenous levels. (**B**) Cells were treated with reversine immediately prior to 4 hr timelapse acquisition, and timing from NEBD to anaphase onset (mitotic exit timing) was collected. Data is representative of four independent experiments,

*Figure 4 continued on next page*

*Figure 4 continued*

n = 23 for each cell line. On average, FLAG-APC4 cells take 48.65 min to mitotic exit (Standard Deviation = SD = 15.99) while reversine treated FLAG-APC4 cells take 34.87 min (SD = 11.00). This difference is significant (p=0.0014). FLAG-APC4$^{KR}$ cells take 81.48 min to mitotic exit (SD = 30.78) while reversine treated FLAG-APC4$^{KR}$ cells take 78.78 min (SD = 21.41). This difference is not statistically significant (p=0.73). Two-tailed t-tests were used to calculate significance. (C–H) Representative cells from timelapse acquisition beginning with NEBD to anaphase onset in cells with or without 50 nM or 1 µM reversine are shown with timestamps indicated in minutes. (I) Activity assays using unmodified or in vitro sumoylated APC/C was performed with recombinant myc-Securin substrate, in the presence or absence of recombinant co-activator Cdc20. (J–K) Immunoblots of immunopurified APC/C using antibodies specific for APC4 and SUMO2/3 with SUMO modification of APC/C complex indicated with '*'. (L) Activity assays using unmodified or in vitro sumoylated APC/C was performed with recombinant myc-Securin substrate, in the presence or absence of recombinant co-activator Cdc20 and inhibitory MCC complex as indicated. (M–O) Immunoblots of immunopurified APC/C using antibodies specific for APC4, APC3 and SUMO2/3, with SUMO modification of APC/C complex indicated with '*'.

DOI: https://doi.org/10.7554/eLife.29539.014

The following videos are available for figure 4:

**Figure 4—video 1.** Movie of Flag-APC4 WT cells treated for 48 hr with siAPC4 and APC4 WT induction with doxycycline.

DOI: https://doi.org/10.7554/eLife.29539.015

**Figure 4—video 2.** Movie of Flag-APC4 WT cells treated for 48 hr with siAPC4 and APC4 WT induction with doxycycline.

DOI: https://doi.org/10.7554/eLife.29539.016

**Figure 4—video 3.** Movie of Flag-APC4$^{KR}$ cells treated for 48 hr with siAPC4 and APC4$^{KR}$ induction with doxycycline.

DOI: https://doi.org/10.7554/eLife.29539.017

**Figure 4—video 4.** Movie of Flag-APC4$^{KR}$ cells treated for 48 hr with siAPC4 and APC4$^{KR}$ induction with doxycycline.

DOI: https://doi.org/10.7554/eLife.29539.018

sumoylation had an effect on MCC-dependent inhibition of APC/C activity (*Figure 4L–O*). Again, indistinguishable results were obtained with control and SUMO-modified APC/C complexes. Thus, sumoylation of APC4 does not directly affect APC/C activity toward securin modification or modulate inhibitory effects of MCC under these assay conditions.

## APC4 sumoylation does not affect SAC protein or APC/C localization at kinetochores

Because the metaphase delay observed in APC4$^{KR}$-expressing cells is dependent on the SAC, we next evaluated whether defects may exist in SAC activation or silencing by assessing the recruitment and release of SAC proteins from kinetochores of aligned and unaligned chromosomes. Mitotic cells were analyzed using antibodies specific for Mad1, Mad2, BubR1, and Cdc20 (*Figure 5A–D*). As expected, in APC4-expressing cells, each of these SAC proteins concentrated to high levels at kinetochores in prometaphase, and these levels decreased significantly following chromosome congression in metaphase. Indistinguishable results were obtained using APC4$^{KR}$-expressing cells, indicating that the activation and silencing of SAC signaling by kinetochores is not dependent on APC4 sumoylation.

APC/C itself also localizes to kinetochores and this localization may be critical for its function (*Acquaviva et al., 2004*; *Jörgensen et al., 1998*; *Topper et al., 2002*). Moreover, sumoylation affects protein localization, including targeting to kinetochores in mitosis (*Joseph et al., 2002*; *Zhang et al., 2008*). To investigate whether the metaphase delay observed in APC4$^{KR}$-expressing cells may result from defects in APC/C localization, we performed immunofluorescence microscopy on FLAG-tagged APC4 and APC4$^{KR}$-expressing cells. Cells were co-stained with anti-FLAG and CREST (a kinetochore and centromere marker) antibodies. Both APC4 and APC4$^{KR}$ were detected at kinetochores in prometaphase and metaphase cells at similar levels (*Figure 5E–F*). In addition, kinetochore localization dissipated similarly in APC4 and APC4$^{KR}$-expressing cells upon anaphase onset, further indicating that sumoylation does not affect the association of APC/C with kinetochores.

## APC2 contains a conserved C-terminal SIM critical for APC/C function

To identify possible underlying mechanisms for how APC4 sumoylation may affect APC/C activation, we analyzed high-resolution cryo-EM structures of the APC/C (*Alfieri et al., 2016*; *Yamaguchi et al., 2016*; *Zhang et al., 2016*). While the extreme C-terminal residues of APC4 (residues 757–808) containing the sumoylation sites are disordered and not visible in these structures, the location of the structured C-terminus (ending in residue 756) suggests that sumoylated APC4 residues are in close proximity (within ~35 Å) to the C-terminus of APC2 (*Figure 6A*). Notably, APC2 is part of the

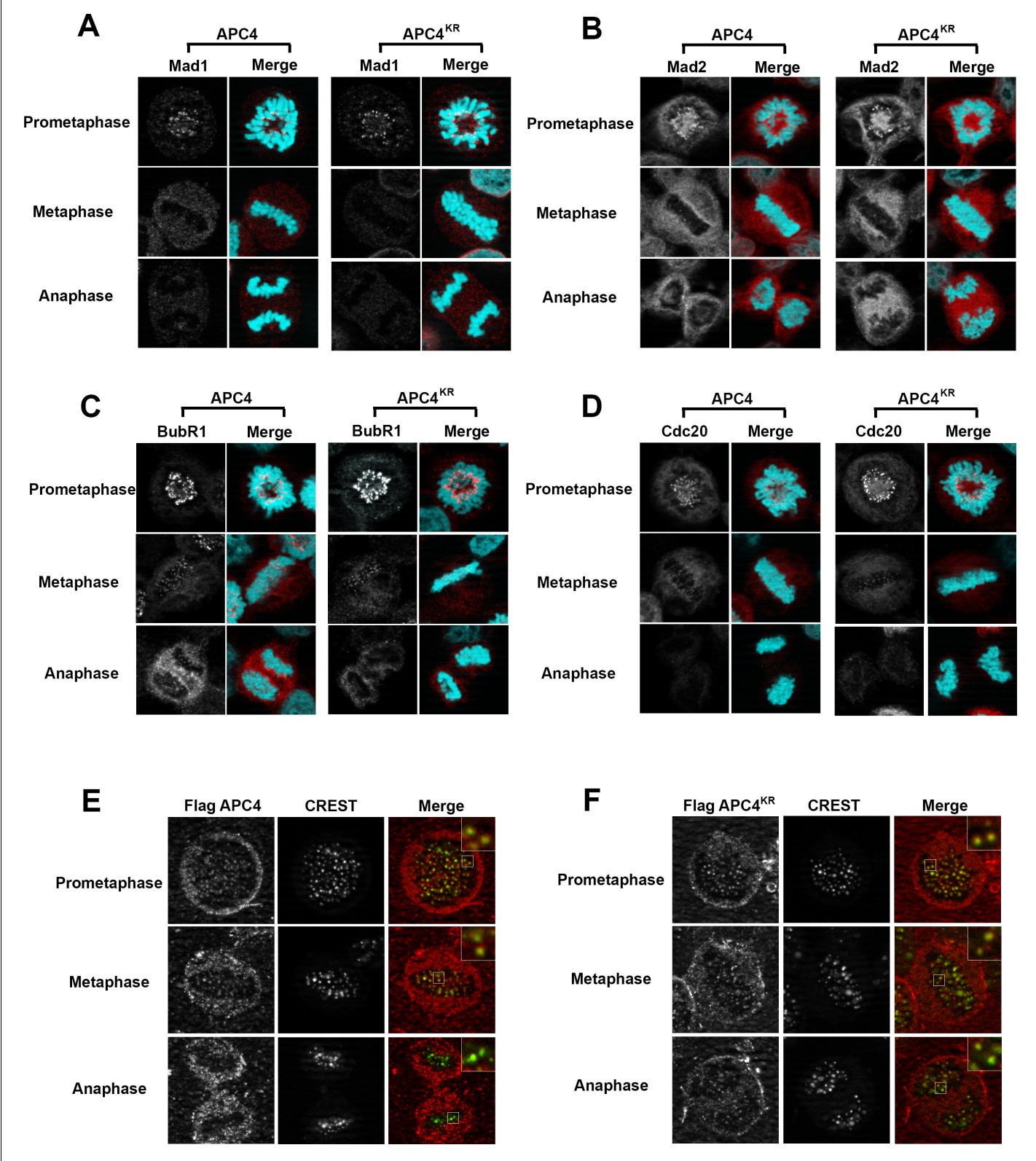

**Figure 5.** APC4 sumoylation does not affect association of APC/C or SAC proteins with kinetochores. Stable inducible cell lines were depleted of endogenous APC4 using siRNA for 48 hr with concomitant induction of FLAG-APC4 or FLAG-APC4$^{KR}$ expression and stained with antibodies specific to (A) Mad1, (B) Mad2, (C) BubR1, and (D) Cdc20. Chromatin (colored in teal) was visualized by detection of YFP-H2B expressed in these cell lines. (E and *Figure 5 continued on next page*

*Figure 5 continued*

F) Inducible cell lines were treated as described above. Cells were stained with FLAG and CREST-specific antibodies and analyzed by immunofluorescence microscopy. Boxed regions are magnified to show co-localization at kinetochores during prometaphase and metaphase.

DOI: https://doi.org/10.7554/eLife.29539.019

structurally dynamic catalytic core of APC/C and critical for E2 binding and positioning (*Brown et al., 2015*; *Yamaguchi et al., 2016*). Sequence analysis of APC2 revealed a potential SUMO interacting motif (SIM) in the C-terminus that is conserved across mammals (*Figure 6B*). The

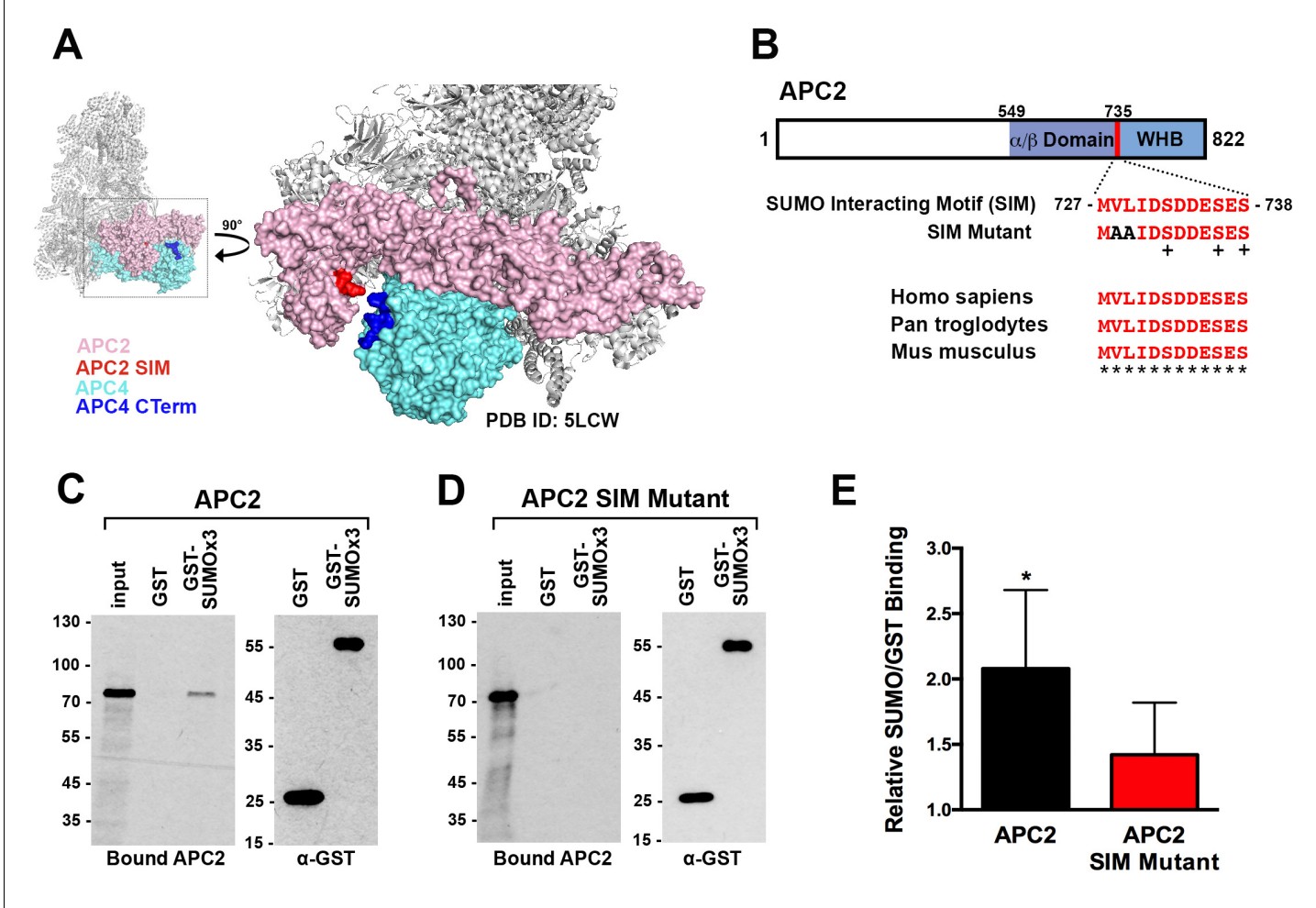

**Figure 6.** APC2 contains a conserved C-terminal SIM. (A) APC2 (pink) contains a predicted SIM (red) near its C-terminus that is in close proximity to the C-terminus (blue) of APC4 (teal). (PDB ID: 5LCW [*Alfieri et al., 2016*]). (B) Schematic depicting the APC2 SIM (red) between residues 727–738 and its conservation in mammals, with sequence homology indicated by '*" and phosphorylation sites indicated by '+". Previously characterized (*Brown et al., 2015*) α/β and WHB domains extend from residues 549–727 and 735–822, respectively. Alanine substitutes at V728 and L729 were used to generate a SIM Mutant. (C and D) APC2 and an APC2 SIM mutant were produced in rabbit reticulocyte lysate in the presence of [$^{35}$S]-methionine. The expressed proteins were incubated with immobilized, recombinant GST or GST-SUMO2 trimer (SUMOx3) and binding was analyzed by SDS-PAGE and autoradiography. (E) Quantitation from three independent binding assays. Values from scintillation counting were used to generate relative binding ratios. Means are plotted and bars represent standard deviations. Two-tailed t-tests were performed: p=0.04 for APC2 wild type binding to GST-SUMOx3/GST, p=0.08 for APC2 SIM Mutant binding to GST-SUMOx3/GST.

DOI: https://doi.org/10.7554/eLife.29539.020

The following figure supplement is available for figure 6:

**Figure supplement 1.** APC2 contains a C-terminal SIM.

DOI: https://doi.org/10.7554/eLife.29539.021

SIM consists of a stretch of hydrophobic residues followed by negatively charged aspartic acid residues from 727 to 738, and is positioned just prior to the winged helix B domain (APC2$^{WHB}$) that is involved in both MCC and UbcH10 binding (*Alfieri et al., 2016*; *Brown et al., 2015*; *Tang et al., 2001*).

To determine whether this predicted SIM mediates interactions between APC2 and SUMO, we expressed wild-type APC2 and a SIM mutant, containing alanine substitutions at residues 728 and 729 (*Figure 6B*), in rabbit reticulocyte lysate supplemented with [$^{35}$S]-methionine. Translated proteins were pulled down with immobilized recombinant GST or a recombinant GST-SUMO fusion protein (GST-SUMO2 × 3), and binding was evaluated by SDS-PAGE and autoradiography as well as scintillation counting. This analysis revealed specific interactions between APC2 and SUMO2 that was dependent on the predicted C-terminal SIM (*Figure 6C–E*). Binding assays performed using deletion mutants further verified the presence of this single functional SIM in the C-terminus of APC2 (*Figure 6—figure supplement 1A–D*).

Based on its predicted proximity to sumoylated APC4 residues, we hypothesized that non-covalent interactions between the C-terminus of APC2 and SUMO may contribute to the observed functional role of APC4 sumoylation during the metaphase-anaphase transition. To test this hypothesis, we generated stable cell lines allowing for inducible expression of FLAG-tagged wild-type APC2 or the APC2 SIM mutant defective in SUMO binding (APC2$^{SM}$) (*Figure 7A*). Endogenous APC2 was depleted by siRNA knockdown in the presence and absence of concomitant transgene expression (*Figure 7B*) and effects on cell cycle progression were evaluated by timelapse microscopy. As expected, depletion of APC2 resulted in severe mitotic defects comparable to those observed in APC4-depleted cells, although defects were only observed in ~50% of cells, presumably due to incomplete knockdown (*Figure 7B and C*, *Figure 7—video 1*). Expression of APC2$^{SM}$ was much less effective than expression of wild-type APC2 in rescuing depletion defects (*Figure 7—videos 1* and *2*). In particular, although all cells progressed from NEBD to metaphase normally, a significant fraction of APC2$^{SM}$-expressing cells exhibited prolonged metaphase delays comparable to those observed in APC4$^{KR}$-expressing cells (*Figure 7D–G*). On average, APC2 cells took ~25 min to progress from NEBD-metaphase and 18 min from metaphase to anaphase, while APC2$^{SM}$ cells took ~24 min to progress from NEBD to metaphase and 35 min from metaphase to anaphase. These findings are consistent with APC2-SUMO interactions facilitating timely anaphase onset.

To further evaluate if SUMO-SIM interactions are important for APC/C function, we established a stable cell line for expression of a Flag-tagged APC4$^{KR}$-SUMO2 fusion protein in which SUMO2 was mutated to prevent SIM recognition (*Figure 8A*). Specifically, positions Q35, F36 and I38 in the second β-strand of SUMO2 which are critical for SIM binding were mutated to alanine (*Figure 8A–B*) (*Hecker et al., 2006*; *Huang et al., 2004*; *Sun et al., 2007*; *Zhu et al., 2008*). Following endogenous APC4 depletion, this transgene was induced to near endogenous levels (*Figure 8C*) and cells were imaged using timelapse microscopy (*Figure 8—video 1*). On average, control cells spent 29 min from NEBD to metaphase, while APC4$^{KR}$-SUMO2$^{QFI}$-expressing cells spent 35 min. This difference was not statistically significant (p=0.097). In contrast, control cells spent 18 min on average transitioning from metaphase to anaphase, while APC4$^{KR}$-SUMO2$^{QFI}$ cells spent 53 min, a statistically significant delay (p=0.0011). Thus, recognition of sumoylated APC4 through a SUMO-SIM interaction is functionally important for timely metaphase-anaphase transition.

## Discussion

The multiple signals and mechanisms controlling APC/C activation and the timing of anaphase onset and mitotic exit are not fully understood. Here, we have shown that sumoylation of the APC4 subunit of the APC/C is required for optimal timing of the metaphase-anaphase transition. We have identified sumoylation sites in the C-terminus of APC4 and a SIM in the cullin-homology domain of APC2 that are both critical for timely APC/C activation and mitotic exit. Although multiple mechanisms may explain the effects of APC4 sumoylation on APC/C function, we favor a model in which interactions between sumoylated APC4 and the SIM in APC2 stabilize a conformation of the catalytic core that relieves inhibition and facilitates E2 binding and substrate ubiquitylation in vivo.

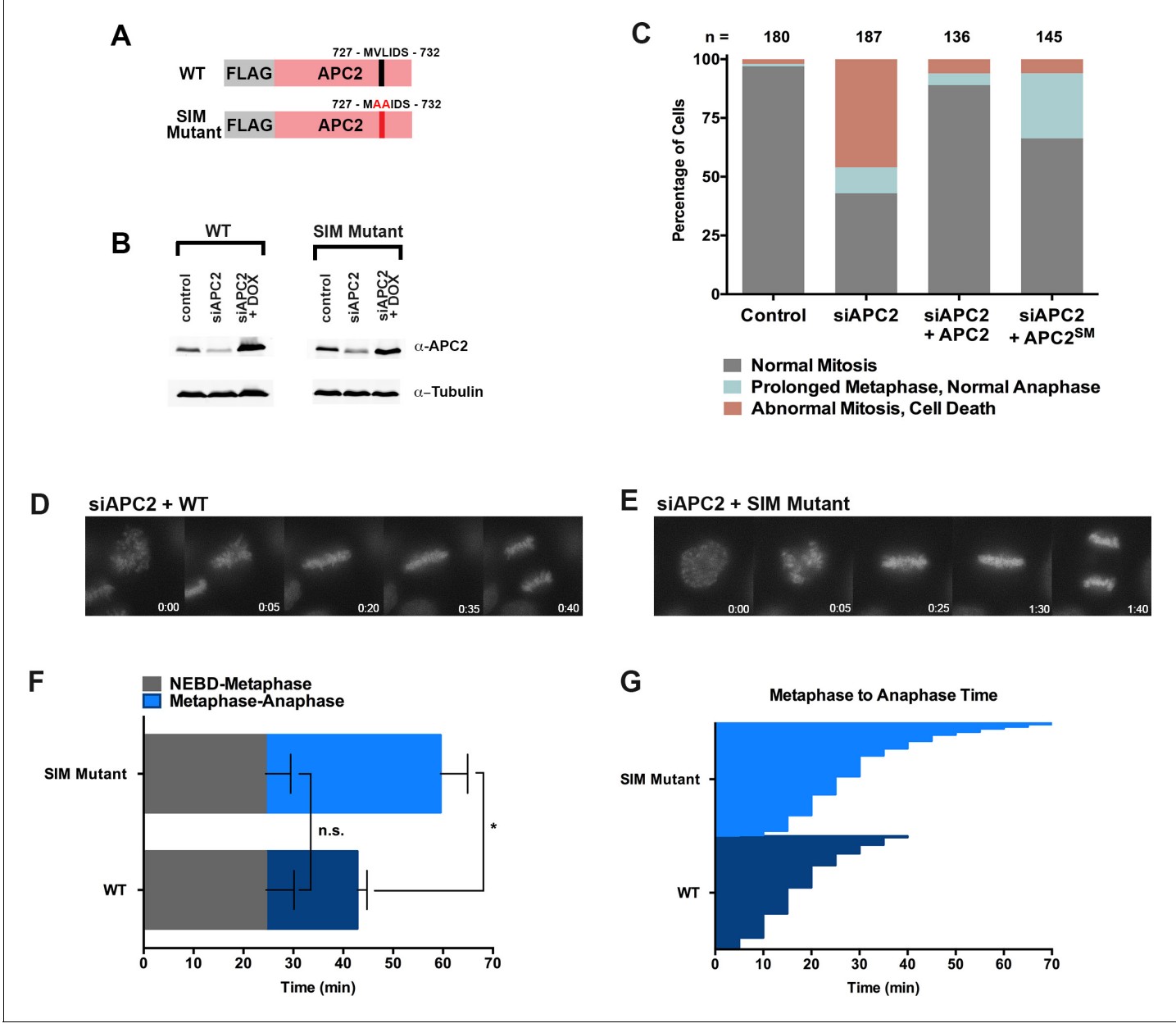

**Figure 7.** The APC2 SIM is required for normal progression from metaphase to anaphase. (**A**) Constructs coding for FLAG-tagged versions of wild-type APC2 or the SIM mutant (APC2[SM]) were used to generate stable inducible cell lines in YFP-H2B HeLa cells. (**B**) Cells were transfected with control or APC2-specific siRNAs and cultured in the presence (+Dox) or absence of doxycycline. Endogenous APC2, FLAG-APC2 and FLAG-APC2[SM] expression levels were analyzed by immunoblot analysis with APC2 and tubulin-specific antibodies. (**C**) Mitotic phenotypes observed in cells treated as in (**B**) are displayed. n > 130 for each cell line. (**D and E**) Representative cells from timelapse acquisition beginning with NEBD to anaphase onset in FLAG-APC2 and FLAG-APC2[SM]-expressing cells with timestamps indicated in minutes. (**F**) Mitotic timing beginning with NEBD to metaphase plate alignment and from metaphase plate alignment to anaphase onset was quantified in FLAG-APC2 and FLAG-APC2[SM] expressing cells. Experiments were performed in triplicate; means are displayed and error bars represent standard deviations. n = 59 for each cell line. Two-tailed t-tests were used to calculate significance: p=0.912 between FLAG-APC2 and FLAG-APC2[SM] timing of NEBD-metaphase, p<0.001 for FLAG-APC2 and FLAG-APC2[SM] timing from metaphase plate alignment to anaphase. (**G**) Individual timing of metaphase to anaphase transition times is displayed for FLAG-APC2 and FLAG-APC2[SM] expressing cells. n = 239 for each cell line.

DOI: https://doi.org/10.7554/eLife.29539.022

The following videos are available for figure 7:

**Figure 7—video 1.** Movie of YFP-H2B (green) HeLa cells treated for 48 hr with siAPC2.

DOI: https://doi.org/10.7554/eLife.29539.023

**Figure 7—video 2.** Movie of FLAG-APC2 WT cells treated for 48 hr with siAPC2 and APC2 WT induction with doxycycline.

*Figure 7 continued on next page*

*Figure 7 continued*

DOI: https://doi.org/10.7554/eLife.29539.024

**Figure 7—video 3.** Movie of FLAG-APC2 SIM mutant (SM) cells treated for 48 hr with siAPC2 and APC2 SM induction with doxycycline.

DOI: https://doi.org/10.7554/eLife.29539.025

## APC4 sumoylation regulates the metaphase-anaphase transition

Several lines of evidence indicate that sumoylation of APC4 is required for timely APC/C activation and mitotic exit. First, cells expressing sumoylation deficient APC4$^{KR}$ underwent normal progression from NEBD to metaphase plate alignment, but exhibited delays during the metaphase-anaphase transition, accompanied by delays in Cyclin B1 degradation. Second, mimicking constitutive APC4 sumoylation through expression of a APC4$^{KR}$-SUMO2 fusion protein (APC4$^{KR-S2}$) rescued the metaphase delay observed in APC4$^{KR}$ cells. Notably, expression of APC4$^{KR-S2}$ had no adverse effects on progression through mitosis, revealing that APC4 sumoylation alone is insufficient to promote premature mitotic exit. This observation suggests that the functional consequences of APC4 sumoylation on APC/C are only manifested following satisfaction of other essential regulatory steps, possibly including checkpoint silencing and other post-translational protein modifications on APC/C.

With regard to checkpoint silencing, our findings revealed that the effect of APC4 sumoylation functions through the SAC. This was evidenced both by the relief of any metaphase delay in cells expressing APC4$^{KR}$ cultured in the presence of high doses of reversine, as well as an insensitivity to low doses of reversine treatment. Analysis of SAC protein localization in mitotic cells indicated that checkpoint signaling by kinetochores is appropriately regulated in APC4$^{KR}$-expressing cells, suggesting that APC4 sumoylation functions downstream of SAC silencing at kinetochores. Exactly how APC4 sumoylation regulates APC/C function, however, remains to be fully understood. We were unable to detect a direct effect of APC4 sumoylation on APC/C activity toward securin using in vitro assays both in the absence and presence of inhibitory MCC complexes. However, it is important to note that these assays were performed using APC/C complexes purified from nocodazole-arrested cells which may not fully replicate conditions at the metaphase-anaphase transition. As noted above, the consequences of APC4 sumoylation on APC/C in vivo appear to require satisfaction of other essential regulatory steps affecting APC/C activation. Of possible significance, the APC2 SIM is phosphorylated at residues 732, 736 and 738 (*Alfieri et al., 2017*) and can be predicted to enhance SUMO binding (*Stehmeier and Muller, 2009*). The timing and consequence of phosphorylation of these residues has not been explored and will be important to investigate in future studies.

## Effects of APC4 sumoylation

Delays in the metaphase-anaphase transition like those observed in APC4$^{KR}$, APC2$^{SM}$, and APC4$^{KR}$-SUMO2$^{QFI}$ expressing cells are unusual and have only been observed in a limited number of cases. Deletion of the spindle and kinetochore-associated (Ska) complex, for example, results in a related anaphase delay. It has been proposed that the Ska complex promotes APC/C localization to chromosomes with the effect of stimulating Cyclin B1 and securin degradation (*Sivakumar et al., 2014*). Although we detected no major defects in APC/C localization to kinetochores in APC4$^{KR}$-expressing cells, it is possible that the chromosome association of APC/C may also be influenced by sumoylation of APC4 specifically by one of the known chromosome-associated SUMO E3 ligases, RanBP2/Nup358 or PIASγ (*Dawlaty et al., 2008*; *Ryu et al., 2010*). Further studies are required to establish whether such a connection exists. Additionally, mutations that inactivate Ube2S, the ubiquitin E2 conjugating enzyme responsible for elongating K11-linked polyubiquitin chains on APC/C substrates, also lead to delays in anaphase onset (*Williamson et al., 2009*). This defect is due in part to inefficient turnover of inhibitory MCC complexes and is checkpoint dependent (*Kelly et al., 2014*).

Also of interest, depletion of the APC15 subunit of the APC/C results in a similar metaphase-anaphase delay observed in APC4$^{KR}$ and APC4$^{SM}$-expressing cells (*Foster and Morgan, 2012*; *Mansfeld et al., 2011*; *Uzunova et al., 2012*). APC15 is not essential for ubiquitylation of CyclinB1 or securin by the APC/C, but instead facilitates turnover of bound MCC complexes mediated in part through auto-ubiquitylation of Cdc20 (*Mansfeld et al., 2011*; *Uzunova et al., 2012*). Cryo-EM analyses of APC/C complexes bound to MCC reveal that APC15 is central to structural changes involved in conversion of the catalytic core from a closed to an open conformation. In the closed

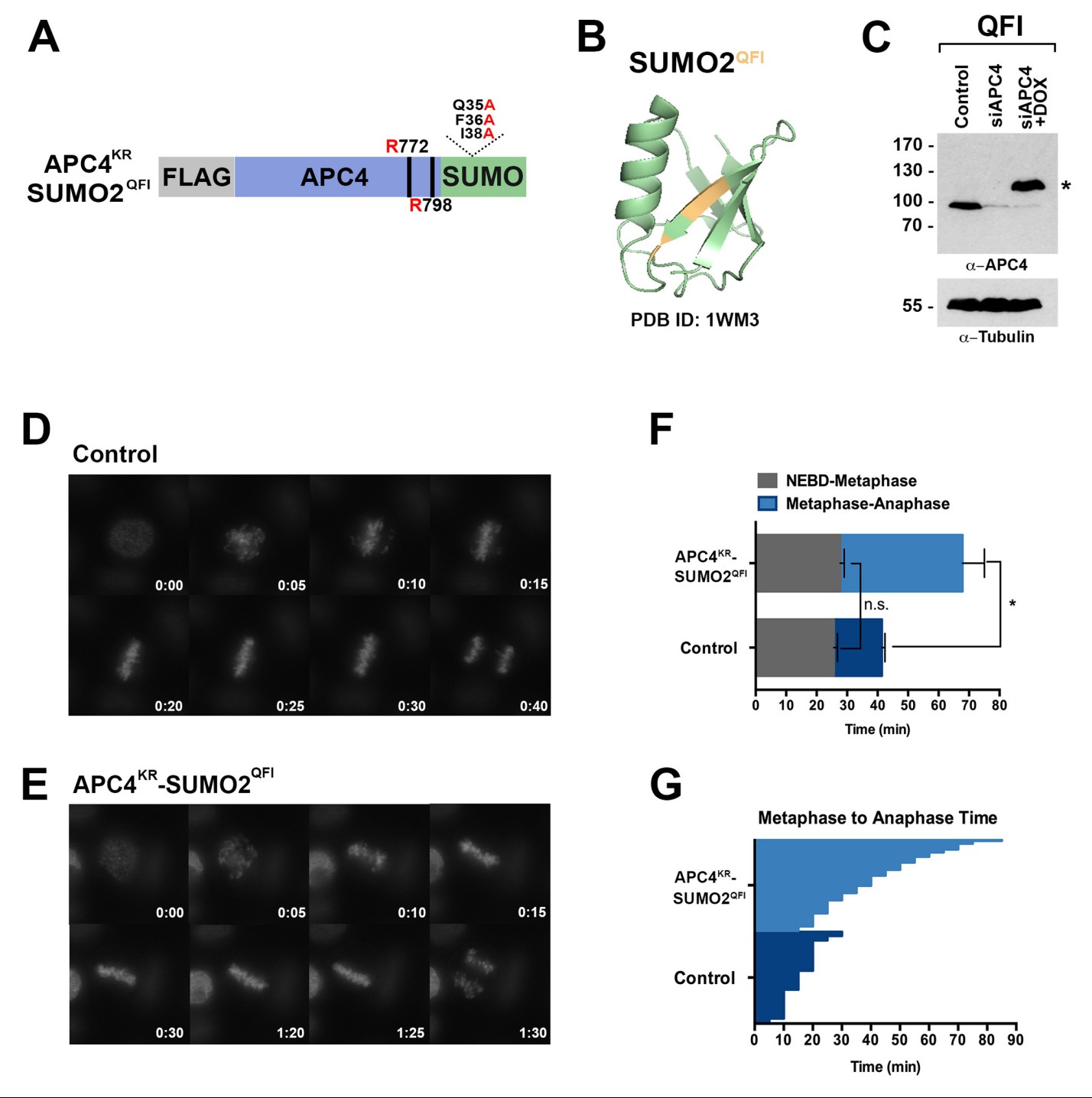

**Figure 8.** SIM binding of APC4[SUMO] is required for normal metaphase-anaphase transitions. (**A**) Construct coding for FLAG-tagged, linear APC4[KR]SUMO deficient of SIM binding (SUMO2-QFI) was used to generate stable inducible cell lines in YFP-H2B HeLa cells. (**B**) The crystal structure of SUMO2 (PDB ID: 1WM3 [*Huang et al., 2004*]) with the SIM binding mutations in the second β-strand (highlighted in yellow) is shown. (**C**) Cells were transfected with control or APC4-targeting siRNAs (Control or siAPC4) and cultured in the presence (+Dox) or absence of doxycycline. Endogenous APC4 and FLAG-APC4[KR]SUMO2[QFI] expression levels were analyzed by immunoblot analysis with APC4 and tubulin-specific antibodies. (**D and E**) Representative cells from timelapse acquisition beginning with NEBD to anaphase onset in control and FLAG-APC4[KR]SUMO2[QFI]-expressing cells with timestamps indicated in minutes. (**F**) Mitotic timing beginning with NEBD to metaphase plate alignment and from metaphase plate alignment to anaphase onset was quantified in control and FLAG-APC4[KR]SUMO2[QFI]-expressing cells. Experiments were performed in triplicate; means are displayed and error bars represent standard deviations (SD). n = 101 for each cell line. Two-tailed t-tests were used to calculate significance: p=0.097 (not

*Figure 8 continued on next page*

*Figure 8 continued*

significant) between Control (average = 28.59 min, SD = 2.19) and FLAG-APC$^{KR}$SUMO2$^{QFI}$ (average = 34.89 min, SD = 1.93) timing of NEBD-metaphase, p=0.0011 for Control (average = 18.28 min, SD = 1.57) and FLAG-APC4$^{KR}$SUMO2$^{QFI}$ (average = 53.15 min, SD = 3.85) timing from metaphase plate alignment to anaphase. (G) Individual timing of metaphase to anaphase transition times is displayed for Control and FLAG-APC4$^{KR}$SUMO2$^{QFI}$-expressing cells. n = 101 for each cell line.

DOI: https://doi.org/10.7554/eLife.29539.026

The following video is available for figure 8:

**Figure 8—video 1.** Movie of FLAG-APC4$^{KR}$-SUMO2$^{QFI}$ cells treated for 48 hr with siAPC4 and SUMO2$^{QFI}$ induction with doxycycline.

DOI: https://doi.org/10.7554/eLife.29539.027

conformation, APC/C is fully inhibited from binding both substrates and the E2 conjugating enzyme UbcH10. In the open conformation, a shift in the MCC complex relieves interactions between the WHB domain of APC2 and BubR1, thereby enabling UbcH10 binding and Cdc20 auto-ubiquitylation. Depletion of APC15 blocks transition to the open conformation and delays progression into anaphase, implying that one mechanism for SAC silencing involves regulation of the closed to open conformation of APC/C and Cdc20 auto-ubiquitylation (*Alfieri et al., 2016*; *Yamaguchi et al., 2016*). Mechanisms affecting the transition between closed and open conformations, however, remain unknown.

Based on the proximity of the SIM in APC2 to the WHB domain, it is tempting to speculate that APC4 sumoylation functions to affect timely anaphase onset at least in part by promoting the transition of APC/C from the closed to open conformation, resulting in permissive E2 binding. Both the APC4 sumoylation sites and the SIM are located within domains that are disordered in currently available structures. Sumoylation could stabilize these flexible domains and a configuration of the WHB domain that facilitates MCC turnover and activation (*Figure 9*). There is precedence for ubiquitin-like modifications influencing the activity of canonical cullin-RING E3 ligases. Specifically, neddylation of the WHB domains of cullins releases the RING domain of Rbx1 from conformational constraints and thereby enhances catalysis and substrate ubiquitylation (*Duda et al., 2008*). Because neddylation of APC2 has yet to be described, it is especially intriguing that sumoylation may also influence APC/C activity by affecting the conformation of the catalytic core through effects on the WHB domain.

## Temporal regulation of APC4 sumoylation

APC4 sumoylation is detectable in all other stages of the cell cycle but peaks during mitosis. The factors controlling APC4 sumoylation levels remain largely uncharacterized but may include SUMO E3 ligases, including RanBP2 and PIASγ. Theses E3 ligases regulate sumoylation of Topo IIα in mitosis and thereby its localization and DNA decatenation activities (*Dawlaty et al., 2008*; *Ryu et al.,*

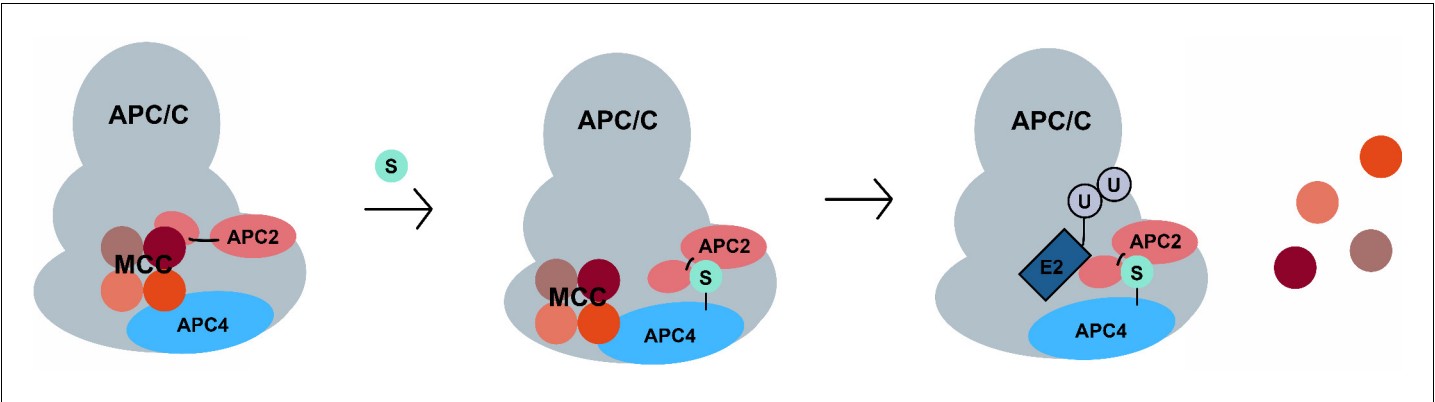

**Figure 9.** A model for SUMO-mediated enhancement of APC/C activity involving APC4 sumoylation and APC2 SUMO binding. SUMO, conjugated to C-terminal lysines in APC4, is proposed to interact with a SIM near the WHB domain of APC2 and stabilize a conformation of the catalytic core that facilitates either MCC turnover, E2 binding or both.

DOI: https://doi.org/10.7554/eLife.29539.028

*2010*). Both RanBP2 and PIASγ localize to centromeres in mitosis and could similarly control APC4 sumoylation in a spatiotemporal manner. Another possible regulator of APC4 sumoylation during mitosis is the SUMO isopeptidase, SENP1. SENP1 is critical for timely progression during the metaphase-anaphase transition (*Cubeñas-Potts et al., 2013*) and our analysis indicates that APC4 is a bona fide SENP1 substrate, as SENP1 depletion results in increased levels of APC4 sumoylation (*Figure 1—figure supplement 1D*). It will therefore be important to further investigate how the activities of SENP1 and possible E3 ligases are coordinated to control APC4 sumoylation in mitosis. Notably, APC4 is also phosphorylated at two residues in close proximity to the C-terminal sumoylation sites (serines 777 and 779) (*Zhang et al., 2016*). Previous evidence for phosphorylation-dependent sumoylation has been described (*Gareau and Lima, 2010*; *Hietakangas et al., 2006*), suggesting that APC4 sumoylation may also be under the regulatory control of mitotic kinases.

## Broader implications and conclusions

Sumoylation is required for the assembly and activities of many cellular structures including ribosomes, PML nuclear bodies, DNA repair foci and kinetochores (*Finkbeiner et al., 2011*; *Jentsch and Psakhye, 2013*; *Matunis et al., 2006*; *Shen et al., 2006*; *Zhang et al., 2008*). In many of these structures, sumoylation functions to promote assembly of protein complexes by facilitating interactions through covalent modification and non-covalent SIM binding. In contrast to these examples, our discovery of SUMO-SIM regulation of the APC/C suggest that sumoylation can also function to modulate the intermolecular interactions and functions of a pre-assembled molecular machine.

In summary, we have shown that both covalent sumoylation and non-covalent SUMO-SIM interactions play important roles in regulating APC/C function and mitotic exit. Although a precise molecular mechanism remains to be determined, we favor a model in which APC4 sumoylation stabilizes a conformation of the catalytic core that facilitates activation and substrate ubiquitylation. A formal test of this model will benefit from further functional studies and structural analysis of sumoylated APC/C. Notably, inhibitors of the SUMO pathway are currently being developed for chemotherapeutic use (*He et al., 2017*), and our studies provide molecular insights into how these inhibitors may function to disrupt cell division.

# Materials and methods

## Plasmids and cell lines

The coding sequence for APC4 was PCR-amplified from pENTR221 ANAPC4 (Ultimate Human ORF Collection, HiT Center, Johns Hopkins University) into pJET 1.2 (ThermoFisher Scientific, Waltham, MA). The linear SUMO2 fusion (without the C-terminal di-glycine) was PCR-amplified and inserted in frame to the C-terminus of APC4 in the vector above. APC2 was subcloned into the pCITE vector from a plasmid (a gift from Hongtao Yu, UT Southwestern). Single N-terminal FLAG tags were inserted using PCR-amplified sequences. Single (APC4$^{K772A}$, APC4$^{K798A}$), double (APC4$^{K772/798A}$, APC4$^{K772/798R}$, APC2$^{SIM\ Mutant}$) and triple mutants (APC4$^{KR}$SUMO2$^{QFI}$) were generated by PCR-based QuickChange site-directed mutagenesis (Agilent) using the wild-type vector as a template.

293FT and YFP-H2B HeLa cells (a gift from Andrew Holland, Johns Hopkins University, Baltimore, MD) were used to generate tetracycline-inducible cell lines. Lentiviruses containing a blasticidin-resistant tetracycline inducible plasmid, in addition to hygromycin-resistant APC4 or APC2 transgenes were incorporated via lentiviral infection as described previously (*Chang et al., 2011*; *Yang and Shih, 2013*). In brief, 1 μg of lentiviral DNA construct, 800 ng psPAX2 (Addgene, Cambridge, MA), and 200 ng pMD2.G (Addgene) were transfected into 293FT cells using XTremeGene HP (Roche) according to manufacturer's instructions. After 48 hr, the viral supernatant was filtered through a 0.44 μM PVDF membrane and transduced to YFP-H2B HeLa cells in the presence of 8 μg/mL polybrene (Santa Cruz Biotechnology). After 24 hr, cells were treated with 10 μg/mL blasticidin (ThermoFisher Scientific) to select for the tetracycline inducible cells for 1 week. Selection for stable cell lines was performed with hygromycin B (Roche) at a final concentration of 200 μg/mL for an additional week concomitant with 10 μg/mL blasticidin selection. Individual blasticidin and hygromycin-resistant colonies were isolated with cloning discs (after ~2 weeks) and tested for APC4 or APC2 expression by immunoblot analysis.

Parent cell lines (293FT, JW36 HeLa, YFP-H2B HeLa and U2OS) were authenticated using STR profiling following *ANSI/ATCC ASN-002–2011, Authentication of Human Cell Lines:Standardization of STR Profiling* guidelines. These cell lines were also tested and verified negative for mycoplasma contamination using a PCR-based MycoDtect[TM] kit from Greiner Bio-One North America, Inc. (Monroe, NC).

## Cell culture and synchronization

293FT, JW36 HeLa, YFP-H2B HeLa, U2OS, and 6xHis-SUMO2 (a gift from Mary Dasso, National Institutes of Health, Bethesda, MD) cells were maintained at 37°C in a 5% $CO_2$ atmosphere. Cells were grown in DMEM medium (Gibco) containing 10% fetal bovine serum (Atlanta Biologicals). Cells were not tested for mycoplasma contamination or authenticated otherwise. Thymidine (Sigma-Aldrich, St. Louis, MO.) was dissolved in DMSO and used at a final concentration of 2 mM; nocodazole (Calbiochem, Burlington, MA) was dissolved in DMSO and used at a final concentration of 100 ng/mL or 3.3 µM as indicated. To synchronize cells into S phase, cells were cultured in the presence of 2 mM thymidine for 19 hr, released in drug-free media for 6 hr, and cultured in 2 mM thymidine for 19 hr. For mitotic timecourse analysis, cells were released from the double thymidine block and harvested in 2x SDS-sample buffer at indicated timepoints.

To weaken SAC signaling, 50 nM of the Mps1 kinase inhibitor, Reversine (Cayman Chemical, Ann Arbor, MI), was used immediately before timelapse acquisition.

## RNAi interference

JW36 HeLa or YFP-H2B HeLa cells were transfected using Lipofectamine RNAiMAX (Invitrogen) and siRNA oligonucleotides at a final concentration of 20 nM for 48 hr and before analysis by immunoblotting, timelapse microscopy, or immunofluorescence microscopy. Stable cell lines were treated with doxycycline to induce expression of FLAG-APC4, FLAG-APC4[KR], or FLAG-APC4[KR]-SUMO-2 concomitant with siRNA depletion of endogenous APC4. Two oligonucleotides targeting the 3' UTR of APC4 were validated, but oligonucleotide two was used for all experiments. APC4 oligonucleotides were purchased from Dharmacon: APC4 Oligo 1: (5' – AGCUUGCCAUUAUUGUGUGUGUAAU – 3'), APC4 Oligo 2: (5' – CAUAGGAGAUGGACUAAGAUGUCUUGG – 3'), scramble control: (5' –C UUCCUCUCUUUCUCUCCCUUGUGA-3'), and SENP1 oligo(b): (5' GCAAAUGGCCAAUGGAGAAA UUCUA-3') and SENP2 oligo(a): (5'AUAUCUGGAUUCUAUGGGAUU-3') were used as previously described (*Cubeñas-Potts et al., 2013*). APC2 oligonucleotides were designed using the BLOCK-iT RNAi Designer (ThermoFisher Scientific) to target the 5' UTR and purchased from Integrated DNA Technologies: 5' – TGGCTGCGCGTGCAGACGTGCGTCA – 3'.

## In vitro sumoylation

APC4 wild-type, APC4[K772A], APC4[K798A], APC4[K772/798A], and RanGAP1 were produced by in vitro transcription and translation in rabbit reticulocyte lysate in the presence of [$^{35}$S]methionine (Perkin Elmer, Waltham, MA) using the TNT T7 Quick Coupled Transcription Translation Kit according to the manufacturer's instructions (Promega, Madison, WI). 2 µL of translation product was added to a 30 µL reaction containing 200 nM human SUMO E1 enzyme, 600 nM human Ubc9, 1.0 µM human SUMO-1 or SUMO-2 proteins, 1 mM ATP, 20 units/mL creatine phosphokinase, 5 mM phosphocreatine, 0.6 mg/ml inorganic pyrophosphatase from *E. coli*, 20 mM HEPES-KOH (pH 7.3), 110 mM potassium acetate, 2 mM magnesium acetate, and 1 mM dithiothreitol (DTT). Reactions were incubated at 30°C for the indicated times and stopped by addition of 2x SDS-sample buffer and analyzed by SDS-PAGE followed by autoradiography. In vitro sumoylation assays are described in more detail at Bio-protocol (*Lee et al., 2018*).

Recombinant SUMO proteins and SUMO enzymes were purified from *E. coli* as previously described (*Yunus and Lima, 2009*).

## Antibody and imaging techniques

For immunoblotting, proteins were separated by SDS-PAGE, transferred onto nitrocellulose membranes, blocked in 5% milk in TS-T, and then probed with the following antibodies diluted in PBS supplemented with 2% BSA and 0.05% $NaN_3$: APC2 (rabbit, a generous gift from Hongtao Yu), APC4 (rabbit, A301-176A, 1:2000, Bethyl Laboratories, Inc), BubR1 (rabbit, GeneTex, 1:1000) Cyclin

B1 (mouse, GNS1: sc-245, 1:200, Santa Cruz Laboratories), Cdc20 (mouse, p55 CDC (E-7): sc-13–162, 1:250, Santa Cruz Laboratories), ECS (DDDDK, goat, A190-101A, Bethyl Laboratories, Inc.), GAPDH (rabbit, TAB1001, OpenBiosystems, 1:15,000), Myc (mouse, a generous gift from Hongtao Yu), SENP1 (rabbit, ab108981, 1:10,000, Abcam), SENP2 (rabbit polyclonal, produced as described previously: (*Goeres et al., 2011*), 1:500), SUMO-1 (21C7, 1:100), SUMO-2/3 (8A2, 1:800), and Tubulin (mouse, DM1A, 1:5000, Sigma-Aldrich. Secondary antibodies conjugated to HRP (Jackson Laboratories) were used at 1:10,000 diluted in 5% milk in TS-T. Immunoblot analysis was performed using either an enzyme-linked chemiluminesence substrate (Luminata Crescendo Western HRP Substrate, EMD Millipore) and developed with film or IR Dye-labeled secondary antibodies (anti-rabbit IgG IRDye 800, LI-COR 926–32211) and imaged using the Odyssey infrared imager (LI-COR). Images were processed with Adobe Photoshop CS6.

To test the localization of Flag-APC4 and Flag-APC4$^{KR}$, cells were depleted of endogenous APC4 using siRNA and induced with doxycycline for 48 hr on glass coverslips. Cells were permeabilized using transport buffer (200 mM HEPES pH 6.5, 110 mM potassium acetate, 20 mM magnesium acetate, 1 µg/mL leupeptin, 1 µg/mL pepstatin A, 20 µg/mL aprotinin, 1 mM phenylmethylsulfonyl fluoride (PMSF), and 20 mM N-ethylmaleimide (NEM), and 20 µg/mL digitonin at room temperature for 15 min. Cells were then washed 1x with PBS, and fixed in 2% formaldehyde at room temperature for 20 min. After washing with PBS, cells were immunostained at room temperature using respective antibodies.

For SAC co-localization studies using immunofluorescence microscopy, stable cell lines were induced with doxycycline to express FLAG-APC4 or FLAG-APC4$^{KR}$ with siRNA to APC4 for 48 hr on glass coverslips. Cells were fixed in 3.5% paraformaldehyde in PBS for 7 min and permeabilized in 0.5% Triton-X 100 in PBS for 20 min at room temperature. Immunostaining was performed with the following antibodies diluted in PBS supplemented with 2% BSA: BubR1 (rabbit, GeneTex, 1:500), Cdc20 (mouse, p55 CDC(E-7): sc13-162, 1:50), CREST (human, 15-235-0001, Antibodies Inc., 1:100), FLAG (mouse, M2, 1:300, Sigma-Aldrich), Mad1 (mouse, 1:500, Active Motif), and Mad2 (rabbit, Covance, 1:500), followed by secondary antibodies to Alexa Fluor 594 and Alexa Fluor 647 (Life Technologies, Grand Island, NY). Images were acquired using a Zeiss Observer Z1 fluorescence microscope with a Zeiss Plan-Apochromat 63x objective (numerical aperture 1.40) and Apotome VH optical sectioning grid (Carl Zeiss, Jena, Germany). Images were obtained at room temperature with immersion oil using a Zeiss AxioCam MRm camera and processed using AxioVision Software Release 4.8.2 and Adobe Photoshop CS6.

For immunoprecipitations to detect SUMO-modified APC4, lysates of YFP-H2B HeLa cells were treated with siRNA oligonucleotides to endogenous APC4 and induced with doxycycline for FLAG-APC4 or FLAG-APC4$^{KR}$ expression for 48 hr. Cell lysates were arrested in prometaphase with a 3.3 µM nocodazole treatment for 4 hr and harvested in RIPA lysis buffer supplemented with protease inhibitors (cOmplete ULTRA EDTA-free tablets, Roche), 1 mM phenylmethylsulfonyl fluoride (PMSF), and 10 mM N-ethylmaleimide, sonicated, and centrifuged 14,000 rpm for 20 min at 4°C. Protein lysates were quantified using a bicinchoninic acid protocol (ThermoScientific) to normalize protein inputs. Cell lysates were incubated with rabbit anti-APC4 antibodies (rabbit, Bethyl Laboratories) immobilized on Protein-A agarose beads (sc-2001, Santa Cruz) and incubated at 4°C overnight. Beads were washed with RIPA buffer 3x and proteins were eluted directly in 2x SDS-sample buffer.

## Ni-NTA affinity purification

To investigate APC4 sumoylation during mitosis, U2OS or 6xHis-SUMO2 U2OS cells were synchronized into S phase using 2 mM thymidine or treated with 100 ng/mL nocodzaole for 16 hr for prometaphase arrest, followed by 2, 4, or 8 hr release into drug-free media. Cells were washed 1x in PBS and flash frozen in liquid nitrogen until assays were performed. Cells were lysed in 6M guanidine HCl lysis buffer, containing: 6M guanidine HCl, 100 mM NaCl, 10 mM imidazole, 10 mM Tris-HCl pH 8.5, and 10 mM β-Mercaptoethanol. Cells were sonicated briefly, cleared by centrifugation, and incubated with pre-washed Ni-NTA agarose (QIAGEN) for 2 hr at 4°C. After a series of wash steps (1x Buffer A: 6M guanidine HCl, 10 mM Tris-HCl pH 8.5, 300 mM NaCl, 20 mM imidazole, 1% TNX-100, 1x Buffer B: 8M urea, 10 mM Tris-HCl pH 8.5, 300 mM NaCl, 20 mM imidazole, 1% TNX-100, 2x Buffer C: 8M urea, 10 mM Tris-HCl pH 6.5, 300 mM NaCl, 20 mM imidazole, 1% TNX-100), proteins were eluted in 2x SDS-sample buffer and analyzed by SDS-PAGE and immunoblotting.

## APC/C ubiquitination assays

For APC/C activity assays, HeLa cells were arrested in prometaphase using overnight nocodazole treatment. Cell pellets were washed 1x with PBS and flash frozen in liquid nitrogen. Ubiquitylation reactions were performed as previously described (*Brulotte et al., 2017*; *Ji et al., 2017*). Briefly, APC/C was isolated from HeLa cell extracts with anti-APC3 antibody coupled Protein A beads (Bio-Rad). After a brief wash, in vitro sumoylation reactions were performed at room temperature for 2 hr using SUMO assay buffer (50 mM HEPES pH 7.7, 100 mM NaCl, 1.5 mM $MgCl_2$, and 1 mM DTT) containing 0.6 µM human SUMO E1 enzyme, 1.5 µM human Ubc9, 1 µM human SUMO2, 1x Energy Mix, 1 mM ATP, 1x IPP (inorganic pyrophosphatase, New England Biolabs), and 1 mM DTT. After a brief wash, beads were incubated with 1 µg recombinant human full-length Cdc20 for 1 hr. Functional MCC was formed by an incubation of BUBR1N, Cdc20, and Mad2 for 20 min as previously described (*Brulotte et al., 2017*). APC/C$^{Cdc20}$ beads were incubated with the pre-formed MCC and washed. The ubiquitination reaction was performed in the presence of securin-Myc using ubiquitylation reaction buffer (10 mM HEPES, pH 7.5, 100 mM KCl, 0.1 mM $CaCl_2$, 1 mM $MgCl_2$) including 150 µM bovine ubiquitin (Sigma), 5 µM Uba1, 750 nM UbcH10, 3 µM Ube2S, and 5 µM Myc-tagged Cyclin B1, supplemented with 1X energy mixture (7.5 mM phosphocreatine, 1 mM ATP, 100 µM EGTA, and 1 mM $MgCl_2$). After incubation at room temperature with gentle shaking, the reactions were quenched with SDS loading buffer and analyzed by Western blotting.

## Timelapse microscopy

For live-cell imaging, cells were cultured in Lab-Tek Chambered #1.0 Borosilicate Coverglass slides (Nunc, Rochester, NY), doxycycline induced and siRNA treated targeting APC4, and then imaged 48 hr post-transfection. Immediately before imaging, cells were switched to pre-warmed $CO_2$-independent media (Invitrogen) supplemented with 10% FBS. Cells were maintained at 37°C on a Zeiss Observer Z1 fluorescence microscope fitted with an incubation chamber. Images were acquired using a Zeiss EC Plan-Neofluar 40x objective (numerical aperture 1.3) every 5 min for 16 hr with a Zeiss AxioCam MRm camera and processed using AxioVision Software Release 4.8.2. Data analysis was performed using Prism six for Mac OS X (GraphPad Software, Inc.).

## Statistical analysis

Statistical analyses were carried out using Prism six for Mac OSX (GraphPad Software, Inc.) with two-tailed student $t$ tests. Data with a p-value<0.05 was considered as statistically significant.

# Acknowledgements

We thank Andrew Holland for YFP-H2B HeLa cells and thoughtful discussions and assistance; Mary Dasso for the 6xHis-SUMO2 U2OS cells and thoughtful discussions and suggestions; Janice Evans for the use of the microscope for timelapse imaging; all of the members of the Matunis Lab and Marie Morrow for their discussions, ideas, and critical reading of the manuscript. We would also like to thank Dr Alfred CO Vergetaal for sharing unpublished data. Work in the Yu laboratory is supported by grants from the Cancer Prevention and Research Institute of Texas (RP120717-P2 and RP160667-P2) and the Welch Foundation (I-1441). HY is an Investigator with the Howard Hughes Medical Institute. This work was funded by National Institutes of Health Grant R01 GM060980 (to MJM) and T32 CA009110 (to CL).

# Additional information

### Funding

| Funder | Grant reference number | Author |
|---|---|---|
| National Institutes of Health | T32 CA009110 | Christine C Lee |
| Howard Hughes Medical Institute | | Hongtao Yu |
| Cancer Prevention and Research Institute of Texas | RP120717-P2 | Hongtao Yu |

| Welch Foundation | I-1441 | Hongtao Yu |
|---|---|---|
| National Institutes of Health | GM060980 | Michael J Matunis |
| Cancer Prevention and Research Institute of Texas | RP160667-P2 | Hongtao Yu |

The funders had no role in study design, data collection and interpretation, or the decision to submit the work for publication.

## Author contributions

Christine C Lee, Conceptualization, Data curation, Formal analysis, Validation, Investigation, Visualization, Methodology, Writing—original draft, Writing—review and editing; Bing Li, Data curation, Investigation, Methodology; Hongtao Yu, Conceptualization, Resources, Formal analysis, Supervision, Writing—review and editing; Michael J Matunis, Conceptualization, Resources, Data curation, Formal analysis, Supervision, Funding acquisition, Writing—original draft, Project administration, Writing—review and editing

## Author ORCIDs

Christine C Lee http://orcid.org/0000-0003-4540-1628
Hongtao Yu http://orcid.org/0000-0002-8861-049X
Michael J Matunis http://orcid.org/0000-0002-9350-6611

## Decision letter and Author response

Decision letter https://doi.org/10.7554/eLife.29539.031
Author response https://doi.org/10.7554/eLife.29539.032

# Additional files

## Supplementary files

• Transparent reporting form
DOI: https://doi.org/10.7554/eLife.29539.029

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
