## [Decision Letter]

Thank you for submitting your article "Sumoylation Promotes Optimal APC/C Activation and Timely Anaphase" for consideration by *eLife*. Your article has been favorably evaluated by Andrea Musacchio (Senior Editor) and three reviewers, one of whom is a member of our Board of Reviewing Editors.

The reviewers have discussed the reviews with one another and the Reviewing Editor has drafted this decision to help you prepare a revised submission.

Summary:

In this manuscript, Lee and co-workers investigate the role of the ubiquitin-like modifier SUMO in controlling the activity of the anaphase promoting complex (APC/C), an essential regulator of cell division. The authors initially validate previously published proteomics data by demonstrating that the APC4 subunit is modified by SUMO at K772 and 798. By using an siRNA-depletion and complementation approach they show that cells expressing a SUMOylation-deficient APC4 mutant (APC4 2KR) exhibit a metaphase to anaphase delay. This phenotype could be rescued by linearly fusing SUMO2 to APC4 2KR supporting the idea that SUMO2 conjugation to APC4 indeed regulates APC activity and proper mitotic progression. In subsequent experiments the authors rule out an involvement of the SAC in this process, but propose that SUMO-mediated binding of APC4 to a SIM in APC2 stimulates APC activity. This model is based on available structural data and the observation that APC2 binds to SUMO via a C-terminal SIM region. Moreover, when expressed in cells in the absence of endogenous APC2 the SIM mutant of APC2 recapitulates the phenotype observed in cells expressing APC2 KR. Based on their findings the authors propose that APC4 SUMOylation stabilizes a conformation that of APC2 that is optimal for E2 enzyme UbcH10 thereby facilitating substrate modification. Altogether this work provides substantial evidence that APC/C activity is controlled by the SUMO system thus expanding previous observations made in lower eukaryotes. In principle, this is an important and significant finding. However, to support publication in *eLife*, the major experiments described below will have to be performed to strengthen the authors' conclusions.

Essential revisions:

1) The authors need to strengthen their evidence showing that APC4 is indeed SUMOylated, and that this SUMO-modification is regulated during the cell cycle. The Western blots in Figure 1 are not sufficient to support this important claim, but instead APC4-IPs and anti-SUMO Westerns, or better, denaturing HIS-SUMO purifications during cell cycle time course would be required.

2) The analysis of in vitro-ubiquitylation activity by the APC/C +/- APC4-SUMOylation needs to be significantly improved (Figure 4). This is a major weakness of the current manuscript. The authors need to compare WT-APC4 with APC4-KR2, and they need to provide quantitative information to the extent of APC/C-stimulation by APC4-SUMOylation. Their low APC/C activity is likely caused by incomplete incorporation of APC4-FLAG into the APC/C, and thus, they either need to integrate a SUMO-tag after the APC4-locus using CRISPR/Cas9, or ensure that FLAG-APC4 is purified only in the context of complete APC/C (for example, by sequential IPs).

3) The authors need to improve their data on showing that APC4 SUMOylation and APC2-SIM function are connected. Without such data, the manuscript would remain at a rather descriptive level. This could be done by asking whether the APC4-KR2-SUMO fusion rescues a APC2-SIM mutant (shouldn't). Also, they need to improve the specificity controls for these experiments, for example by showing that a SUMO mutant that disrupts recognition by SIMs can't be recognized by the SIM of APC2 and also fails to rescue the APC4-KR2 phenotype (this would at least implicate SUMO recognition, rather than steric effects, in the function of APC4 SUMOylation).

---

## [Author Response]

Essential revisions:1) The authors need to strengthen their evidence showing that APC4 is indeed SUMOylated, and that this SUMO-modification is regulated during the cell cycle. The Western blots in Figure 1 are not sufficient to support this important claim, but instead APC4-IPs and anti-SUMO Westerns, or better, denaturing HIS-SUMO purifications during cell cycle time course would be required.

It was suggested that more conclusive evidence was required to demonstrate that APC4 is indeed sumoylated and that sumoylation is regulated during the cell cycle. We have produced several lines of evidence demonstrating the sumoylation of APC4. First, we have previously published immunopurification and western blotting data showing the modification of endogenous APC4 by both SUMO-1 and SUMO-2/3 (Cubeñas-Potts et al., 2015). In addition, immunopurifications of Flag-tagged wild type and mutant APC4 and anti-SUMO westerns are included in Figures 1F-G and Figure 1—figure supplement 1C of the current study. As recommended, we also now demonstrate more conclusively that APC4 sumoylation is cell cycle regulated by including His-SUMO purification and time course analysis in the newly added Figure 1—figure supplement 1A.

2) The analysis of in vitro-ubiquitylation activity by the APC/C +/- APC4-SUMOylation needs to be significantly improved (Figure 4). This is a major weakness of the current manuscript. The authors need to compare WT-APC4 with APC4-KR2, and they need to provide quantitative information to the extent of APC/C-stimulation by APC4-SUMOylation. Their low APC/C activity is likely caused by incomplete incorporation of APC4-FLAG into the APC/C, and thus, they either need to integrate a SUMO-tag after the APC4-locus using CRISPR/Cas9, or ensure that FLAG-APC4 is purified only in the context of complete APC/C (for example, by sequential IPs).

It was felt that in vitro ubiquitylation activity assays represented a major weakness of the original submission and needed to be significantly improved. We have attempted to address this concern using multiple approaches. As suggested, we attempted sequential IPs and activity assays using our established cell lines expressing Flag-tagged wild type and mutant APC4 but were unable to overcome technical challenges using this approach. We were also unable to establish tagged APC4 cell lines using CRSPR/Cas9. We therefore turned to activity assays using APC/C complexes immunopurified from nocodazole-arrested cells and subsequently sumoylated in vitro prior to analysis. As documented by immunoblotting with APC4 and SUMO specific antibodies, APC4 was efficiently and uniquely modified under our reaction conditions (new Figures 4J-O). Using securin as substrate, we were nonetheless unable to detect any differences in ubiquitylation activity between control APC/C complexes with unmodified APC4 and complexes containing fully sumoylated APC4 (new Figure 4I). We also investigated the ability of APC4 sumoylation to affect MCC binding and inhibition of APC/C activity but again detected no differences in comparison to unmodified APC/C complexes (new Figure 4L).

These new findings have been incorporated into the revised manuscript. Interpretations and possible caveats are presented in the Discussion and take into account multiple and complimentary lines of in vivo evidence supporting an important role for APC4 sumoylation in affecting APC/C activation or activity. Although the precise mechanism by which APC4 sumoylation affects APC/C remains to be determined, we hope the reviewers agree that our findings are important and provide a valuable foundation for future biochemical and structural studies.

3) The authors need to improve their data on showing that APC4 SUMOylation and APC2-SIM function are connected. Without such data, the manuscript would remain at a rather descriptive level. This could be done by asking whether the APC4-KR2-SUMO fusion rescues a APC2-SIM mutant (shouldn't). Also, they need to improve the specificity controls for these experiments, for example by showing that a SUMO mutant that disrupts recognition by SIMs can't be recognized by the SIM of APC2 and also fails to rescue the APC4-KR2 phenotype (this would at least implicate SUMO recognition, rather than steric effects, in the function of APC4 SUMOylation).

It was requested that experimental evidence connecting APC4 sumoylation and functional requirements for the SIM in APC2 be strengthened. As recommended, we investigated whether fusing a SUMO mutant defective in SIM binding to APC4^KR^ would rescue mitotic defects of APC4^KR^ expressing cells. As illustrated in new Figure 8, SIM recognition by SUMO fused to APC4^KR^ is required for rescue of mitotic defects, demonstrating that SUMO-SIM interactions are indeed critical for the effect of sumoylation on APC/C.